# Near-Optimal Online Deployment and Routing for Streaming LLMs

**Shaoang Li, Jian Li**
Stony Brook University

## Abstract

The rapid pace at which new large language models (LLMs) appear, and older ones become obsolete, forces providers to manage a streaming inventory under a strict concurrency cap and per-query cost budgets. We cast this as an online decision problem that couples *stage-wise deployment* (at fixed maintenance windows) with *per-query routing* among live models. We introduce `StageRoute`, a hierarchical algorithm that (i) optimistically selects up to $M_{\max}$ models for the next stage using reward upper-confidence and cost lower-confidence bounds, and (ii) routes each incoming query by solving a budget- and throughput-constrained bandit subproblem over the deployed set. We prove a regret of $\tilde{\mathcal{O}}(T^{2/3})$ with a matching lower bound, establishing near-optimality, and validate the theory empirically: `StageRoute` tracks a strong oracle under tight budgets across diverse workloads.

## 1 Introduction

The proliferation of LLMs has transformed a broad array of applications, delivering unprecedented advances in natural-language understanding and generation (Radford et al., 2019; Brown et al., 2020; Wang et al., 2023b; OpenAI, 2023; Chowdhery et al., 2023; Touvron et al., 2023). Yet LLMs differ markedly in both performance and cost: some offer state-of-the-art capabilities at a premium, while others are more affordable but less effective. Practitioners therefore face a continual accuracy-expenditure tradeoff when deciding which models to operate and when to use them. This has motivated *LLM routing* (Ding et al., 2024; Hu et al., 2024), where a system chooses, query by query, which model to invoke to maximize task quality under cost constraints. However, focusing solely on per-query routing overlooks a more fundamental decision that precedes it: **which models are deployed at all.**

In practice, the operational landscape is unusually fluid. New models arrive continuously with distinct accuracy, latency, and pricing profile (Feng et al., 2025), while production systems must respect hard limits such as rate ceilings and deployment quotas. For example, Azure OpenAI Service caps each resource at 32 standard and 5 fine-tuned deployments by default, and enforces model-specific rate ceilings (e.g., for GPT-4.1: 1,000 requests per minute (RPM) and 1M tokens per minute (TPM)) (Microsoft Azure, 2025). This confluence of a dynamic model pool and strict operational caps recasts the problem into two timescales: a slower *stage-wise* deployment process that decides which models stay alive under a concurrency cap, and a faster *per-query* routing process that assigns each request among the currently deployed models while meeting budget and throughput constraints. *The deployment choice is foundational, since it determines the entire action space for any routing policy.* Table 1 maps recent LLM routing systems to three axes and highlights a gap in current approaches.

We study this setting as *an online decision problem* that couples two intertwined choices (Figure 1): (1) *stage-wise deployment* at fixed update points, where the operator decides which models to deploy for the next stage subject to a hard concurrency cap and deployment costs. This high-stakes decision defines the *action set* for the subsequent execution of (2) *per-query routing*, where each incoming query is sent to one of the currently deployed models to maximize quality while obeying a long-term cost budget and per-model throughput limits. Unlike approaches that assume a static model pool or rely on fully offline retraining, our framework admits streaming arrivals of new LLMs and enforces *active-set replacement*: admitting a newcomer may require evicting an incumbent for the rest of the stage. This mirrors real service constraints while enabling continual adaptation.

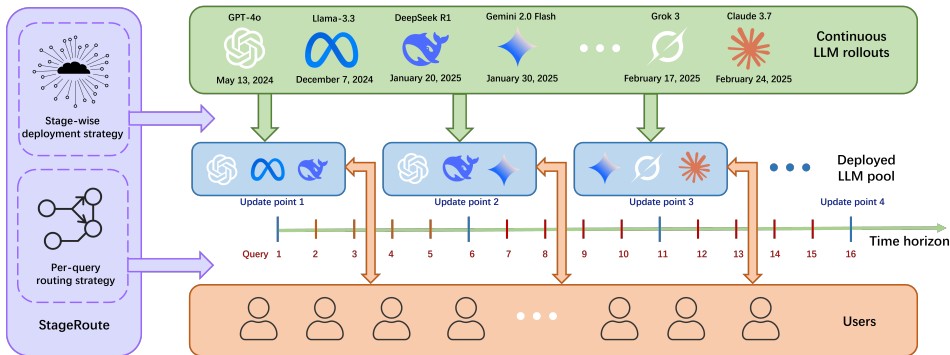

Figure 1: The StageRoute workflow. Newly released LLMs (green) continually enter the candidate pool. At each scheduled update point, StageRoute deploys up to $M_{\max}$ models (blue). Between updates, each query is routed among the current deployment (orange). This two-level loop assimilates fresh models, enforces cost/throughput constraints, and adapts routing in real time.

Three features make this problem technically distinct. First, the hard concurrent-deployment cap induces an *irreversible* exploration-exploitation tradeoff: activating an uncertain model can mean dropping a known, reliable one for an entire stage. Second, decisions occur on two timescales: infrequent, strategic deployment choices constrain frequent, tactical per-query routing. Third, the system must jointly respect a long-term cost budget and per-model throughput limits while selecting a small operational subset under uncertainty. Classical multi-armed bandits (MAB), budgeted formulations (BwK), combinatorial bandits (CMAB), and standard streaming bandits each capture *parts* of this picture, but none natively address *the combination of dynamic availability, staged commitment, and an explicit concurrency cap on the active set.*

To address these challenges, we introduce StageRoute, an algorithm that mirrors the problem's hierarchy (Figure 1). At each update boundary, a *deployment phase* selects the active set for the next stage using optimistic estimates of model quality (UCB) and conservative estimates of cost (LCB), honoring the global budget, per-model throughput limits, and the $M_{\max}$ concurrency cap. Within the stage, each query triggers a *routing phase*: a linear program over the *currently deployment* returns a distribution that maximizes estimated quality under the same constraints, and the query is dispatched by sampling accordingly. *This two-level loop links strategic deployment to fine-grained, adaptive routing, allowing the system to assimilate new information both across and within stages.*[1]

Our contributions in this paper are summarized as follows:

• **Problem formulation.** To our knowledge, we are the first to formalize the *online LLM deployment and routing problem with streaming arrivals*, explicitly modeling a hard concurrency cap, one-time deployment costs, per-model throughput limits, and a long-term cost budget, with stage-level commitment and per-query routing.

• **Algorithm.** We introduce StageRoute, which (i) selects an active set at each update using optimistic performance (UCB) and conservative cost (LCB) estimates under the budget, throughput limits, and the $M_{\max}$ concurrency cap, and (ii) routes each query by solving a budget–throughput LP over the *currently deployed* models. *The design is modular: the routing step can incorporate contextual estimators when features are available, while the deployment step remains unchanged; throughput limits naturally throttle load to mitigate latency spikes.*

• **Theoretical guarantees.** We prove a regret bound of $\widetilde{\mathcal{O}}\big(\sqrt{M_{\max}KT}\big) + \widetilde{\mathcal{O}}\big(NT/(M_{\max}K)\big)$, where $T, K, M_{\max}, N$ are the numbers of queries, update stages, the concurrency cap and arriving models, respectively. The first term captures the statistical learning cost of routing within the deployed set; it grows with the number of active models $M_{\max}$, stages $K$, and horizon $T$. The second term is a structural *model-discovery bottleneck* that quantifies the difficulty of discovering strong newcomers when only $M_{\max}$ models can be live across $K$ stages as $N$ models arrive. Balancing the

---

[1]Relative to nearby bandit frameworks: static-pool routing assumes fixed arms; BwK models consumable budgets but not stage-level active-set replacement; CMAB selects superarms from a fixed base set without streaming arrivals or stage commitment; streaming bandits allow arrivals but do not couple stage-level support selection with per-query routing under both a budget and per-model capacity. See Appendix A for more details.

Table 1: Comparison of LLM routing frameworks. `StageRoute` is the first to address the full real-world setting: a dynamic model pool with streaming arrivals, paired with dynamic stage-wise deployment under a concurrency cap ($M_{\max}$) and cost- and throughput-aware routing.

| Approach | Streaming LLM Models | Dynamic Deployment (with $M_{\max}$ cap) | Cost & Budget Aware | Throughput Limits | Source |
|---|---|---|---|---|---|
| LLM-Blender | – | – | – | – | Jiang et al. (2023) |
| AutoMix | – | – | – | – | Aggarwal et al. (2024) |
| Hybrid-LLM | – | – | – | – | Ding et al. (2024) |
| Zooter | – | – | – | – | Lu et al. (2024) |
| RouterDC | – | – | – | – | Chen et al. (2024b) |
| TensorOpera Router | – | – | ✓ | – | Stripelis et al. (2024) |
| RouteLLM | – | – | ✓ | – | Ong et al. (2025) |
| MESS+ | – | – | ✓ | – | Woisetschläger et al. (2025) |
| UniRoute | ✓ | – | ✓ | – | Jitkrittum et al. (2025) |
| CSCR | ✓ | – | ✓ | – | Shirkavand et al. (2025) |
| `StageRoute` **(ours)** | ✓ | ✓ | ✓ | ✓ | This paper |

two yields a near-optimal $\widetilde{\mathcal{O}}(T^{2/3})$ order, and we give a matching $\Omega(T^{2/3})$ lower bound via a staged-arrival construction. Analytically, we use LP-duality sensitivity with an explicit *support* (active-set) constraint and a regret decomposition that separates routing and deployment across stages.

• **Empirical evaluation.** Simulations show `StageRoute` tracks a strong oracle under tight budgets and is robust across key parameters. We evaluate on *true per-query* scores and costs (RouterBench) across diverse queries, tasks, and languages, demonstrating effectiveness in realistic settings.

## 2 SYSTEM MODEL

We study an *online LLM routing framework*: at each round $t$, a query arrives and must be routed to a suitable LLM. New models may appear at any time, yet they can be *activated* only at discrete deployment intervals (Figure 1). We describe each component of the model in detail.

**Rounds (Queries).** Let $T$ denote the total number of user queries, indexed by $[T] = \{1, 2, \ldots, T\}$. At each time step $t \in [T]$, a query arrives and is immediately routed to an LLM $m_t$ chosen from the currently deployed models according to the algorithm's policy.

**LLM Pool and Deployment Schedule.** Let $\mathcal{M}_t$ be the set of all LLM models that exist and could, in principle, be deployed by time $t$. Each model $m$ has an *availability time* $t_m$; hence $m \in \mathcal{M}_t$ exactly when $t \geq t_m$. Deployment changes occur only at discrete intervals, a schedule determined by the algorithm. The algorithm partitions the time horizon $T$ into $K$ equal-length stages, where $K$ is a tunable hyperparameter of the algorithm. Each stage thus consists of $T/K$ rounds (assuming $T$ is divisible by $K$). The start of stage $k$ is $\tau_k = (k-1)T/K + 1, k = 1, \ldots, K$. Let $M_{\max}$ be the *concurrency cap*, i.e., the maximum number of models that can be deployed simultaneously. At each update point $\tau_k$, the algorithm $\mathcal{A}$ selects a deployed set $\mathcal{D}_k(\mathcal{A}) \subseteq \mathcal{M}_{\tau_k}$ such that $|\mathcal{D}_k(\mathcal{A})| \leq M_{\max}$, which then remains fixed for all $t \in [\tau_k, \tau_{k+1})$. Queries arriving during that stage must be routed to models in the active set $\mathcal{D}_k(\mathcal{A})$. Thus $\mathcal{M}_t$ is *the available pool* at time $t$, while $\mathcal{D}_k(\mathcal{A})$ is *the active subset* that can actually serve queries during stage $k$.

**Operational Performance and Cost.** An LLM's per-prompt quality varies with the input. However, over a long time horizon, it can be reasonably modeled as a random variable centered around a stable mean (Ding et al., 2024). Formally, each model $m$ has an unknown performance distribution $\nu_m(\cdot)$ supported on $[0, 1]$. When $m$ is selected at time $t$, the observed score $r_t \in [0, 1]$ is drawn from $\nu_m(\cdot)$ with mean $\mu_m = \mathbb{E}_{x \sim \nu_m(\cdot)}[x]$. Invoking model $m$ on a query also incurs an *operational cost* $c_{m_t}$ that combines: (i) *Input Cost*: $c_{m_t}^{(\mathrm{in})} = \big(\#$ tokens of input at time $t\big) \times p_{\mathrm{in}}$, where $p_{\mathrm{in}}$ is the per-token input price; (ii) *Output Cost*: $c_{m_t}^{(\mathrm{out})} = \big(\#$tokens of response by $m$ for query $t\big) \times p_{\mathrm{out}}$, with output length drawn from a model-specific distribution $\xi_m(\cdot)$ and unit price $p_{\mathrm{out}}$.

Thus the total cost is $c_{m_t} = c_{m_t}^{(\mathrm{in})} + c_{m_t}^{(\mathrm{out})}$. Because the output token count $c_{m_t}^{(\mathrm{out})}$ depends on the specific model and query and is sampled from $\xi_m(\cdot)$, the total operational cost $c_{m_t}$ for a query handled by model $m_t$ is itself a *random variable*. Operational cost $c_{m_t}$ is inherently bounded by per-token pricing and practical limits on sequence length and generation (e.g., context-window and token caps). Hence we assume $c_{m_t} \in [c_1, c_2]$ for known constants $0 < c_1 \leq c_2 < \infty$.

**Constraints: Budget and Throughput.** When a query arrives at time $t$, with stage index $k$ such that $\tau_k \leq t < \tau_{k+1}$, the algorithm $\mathcal{A}$ selects a model $m_t \in \mathcal{D}_k(\mathcal{A})$. It then observes the reward $r_t \sim \nu_{m_t}(\cdot)$ and incurs cost $c_{m_t}$. The goal is to maximize average reward subject to two main constraints:

(1) *Budget constraint:* $\mathbb{E}\big[\frac{1}{T}\sum_{t=1}^{T} c_{m_t}\big] \leq b + o(1)$. We use an average cost constraint instead of a hard budget constraint because for long-running systems, it provides a degree of flexibility. From a theoretical analysis perspective, since the estimation errors for both performance and cost are governed by the same concentration inequalities, the expected budget violation is upper-bounded by the order of the performance regret. This makes the average cost constraint asymptotically equivalent to a hard constraint for a near-optimal algorithm.

(2) *Per-model throughput limit:* For each deployed LLM model $m$, we specify a throughput limit $\alpha_m$. Let $p_t(m)$ be the probability that the routing policy assigns the query at time $t$ to model $m$. While $t$ lies in stage $k$, the policy must satisfy $p_t(m) \leq \alpha_m, \forall m \in \mathcal{D}_k(\mathcal{A})$. This constraint caps the instantaneous load share each model may receive. For a deterministic decision $m_t$, where $p_t(m_t) = 1$, the requirement reduces to $\alpha_{m_t} \geq 1$. Such limits reflect real-world restrictions like API rate limits (RPM/TPM), bandwidth, or licensing, preventing any single model from being overwhelmed. Hence the deployed set's aggregate throughput must be sufficient to serve every arrival, a condition we formalize next.

**Assumption 1** (Feasibility). *The constraints are feasible. At every time $t$, there exists a subset $\mathcal{S} \subseteq \mathcal{M}_t$ with $|\mathcal{S}| \leq M_{\max}$ such that $\sum_{m \in \mathcal{S}} \alpha_m \geq 1$. The budget $b$ is also large enough to admit a non-trivial routing policy. When required, we assume Slater's condition holds, guaranteeing strong duality for the associated optimization problems.*

**Performance Maximization and Regret.** The goal is to maximize the *expected cumulative performance*, $\mathbb{E}[\sum_{t=1}^{T} \mu_{m_t}]$ subject to: (i) *Deployment Choice:* At each update $\tau_k$, select a deployed set $\mathcal{D}_k(\mathcal{A}) \subseteq \mathcal{M}_{\tau_k}$ with $|\mathcal{D}_k(\mathcal{A})| \leq M_{\max}$; (ii) *Model Selection:* For $t \in [\tau_k, \tau_{k+1})$, choose $m_t \in \mathcal{D}_k(\mathcal{A})$ using probabilities $p_t(m)$ that sum to 1; (iii) *Throughput Constraint:* Ensure $p_t(m) \leq \alpha_m$ for every deployed model $m$; and (iv) *Budget Constraint:* Maintain $\mathbb{E}\big[\frac{1}{T}\sum_{t=1}^{T} c_{m_t}\big] \leq b + o(1)$.

We measure the online policy's performance against an optimal offline benchmark. The foundation of this benchmark is the *Optimal Performance Rate Function*, $V(b, \mathcal{S})$. Given any candidate model set $\mathcal{S}$ and per-query budget $b$, let $V(b, \mathcal{S})$ denote the maximum expected reward per query, which serves as an upper bound for any algorithm operating under these constraints:

$$V(b, \mathcal{S}) = \max_{p \in \Delta(\mathcal{S})} \Big\{ \sum_{m \in \mathcal{S}} \mu_m p(m) \ \Big| \ \sum_{m \in \mathcal{S}} \mathbb{E}[c_m] p(m) \leq b, \sum_{m \in \mathcal{S}} p(m) = 1,$$
$$0 \leq p(m) \leq \alpha_m \text{ for } m \in \mathcal{S}, |\text{supp}(p)| \leq M_{\max} \Big\}. \tag{1}$$

Here, $\Delta(\mathcal{S})$ is the set of probability distributions over $\mathcal{S}$; $p(m)$ is the probability of selecting model $m$; and $\mathbb{E}[c_m]$ is its expected cost. The support constraint $|\text{supp}(p)| \leq M_{\max}$ limits the number of models with positive probability to $M_{\max}$, capturing the combinatorial selection of the best $M_{\max}$ models to use for routing from the entire available pool $\mathcal{M}_{\tau_k}$. If no feasible distribution exists or if $(\mathcal{S} = \varnothing)$, we set $V(b, \mathcal{S}) = 0$.

The *time-varying offline optimum* is $\text{OPT}^* = \sum_{k=1}^{K} (\tau_{k+1} - \tau_k) \cdot V(b, \mathcal{M}_{\tau_k})$, where $(\tau_{k+1} - \tau_k)$ is the length (number of queries) of stage $k$. The regret of an online policy $\mathcal{A}$ is

$$\text{Regret}(\mathcal{A}) = \text{OPT}^* - \mathbb{E}\Big[\sum_{t=1}^{T} \mu_{m_t}\Big], \tag{2}$$

i.e., the expected performance gap between $\mathcal{A}$ and the clairvoyant benchmark, where the expectation is over the algorithm's random choices and outcome variability.

## 3 STAGEROUTE: STAGE-BASED LLM DEPLOYMENT AND ROUTING

We introduce StageRoute (Algorithm 1), a two-level hierarchy that unifies deployment and per-query routing in one algorithm: (i) *Strategic layer.* At each discrete update point $\tau_k$, the algorithm

---

**Algorithm 1** `StageRoute`: stage-based LLM deployment (active-set selection) and online query routing

---

**Require:** Update points $\{\tau_1, \ldots, \tau_K\}$; budget $b$; concurrency cap $M_{\max}$
 1: **Initialize:** Prior parameter estimates; $\tau_0 \leftarrow 0$; $\mathcal{D}_0(\mathcal{A}) \leftarrow \emptyset$
 2: //**Stage-wise Deployment Phase:**
 3: **for** $k = 1$ to $K$ **do**
 4:    Incorporate newly available models $\{m \mid t_m \leq \tau_k < t_m + (\tau_k - \tau_{k-1})\}$; initialize their parameters
 5:    Solve `DeployOPT` (3) for $d^*$ and set $\mathcal{D}_k(\mathcal{A}) \leftarrow \{m \mid d_m^* > 0\}$
 6:    //**Per-query Routing Phase (for query at time $t$):**
 7:    **for** $t = \tau_k$ to $\tau_{k+1} - 1$ (or to $T$ if $k = K$) **do**
 8:       Compute routing distribution $p_t^*$ by solving `RouteOPT` (7)
 9:       Sample $m_t \sim p_t^*$ and route the query to it
10:       Observe reward $r_t$ and cost $c_t$; update statistics for $m_t$
11:    **end for**
12: **end for**

---

decides which models to deploy, adapting to newly available LLMs while respecting the budget and operational constraints. (ii) *Tactical layer.* Between updates, it routes every incoming query in real time among the currently deployed models. The system starts with prior parameter estimates, an empty unexplored-model list, $\tau_0 = 0$, and an empty initial deployment $\mathcal{D}_0(\mathcal{A})$. It then proceeds through $K$ stages, and at the start of each stage $k$ ($t = \tau_k$), the algorithm executes two phases in sequence: **model deployment** followed by **request routing**.

**Model Deployment Phase (Stage Start).** At each update point $\tau_k$, `StageRoute` first incorporates any newly available models (those with $\tau_{k-1} < t_m \leq \tau_k$) and initializes their parameter estimates. With the enlarged pool $\mathcal{M}_{\tau_k}$, `StageRoute` solves the deployment optimization in Eq. (3) to pick the models for stage $k$:

$$\texttt{DeployOPT:} \quad \max_{d \in \Delta(\mathcal{M}_{\tau_k})} \left\{ \sum_{m \in \mathcal{M}_{\tau_k}} \mu_m^U d_m \,\Big|\, \sum_{m \in \mathcal{M}_{\tau_k}} c_m^L d_m \leq b, \sum_{m \in \mathcal{M}_{\tau_k}} d_m = 1, \right.$$
$$\left. 0 \leq d_m \leq \alpha_m \text{ for } m \in \mathcal{M}_{\tau_k}, |\mathrm{supp}(d)| = \min(M_{\max}, |\mathcal{M}_{\tau_k}|) \right\}. \tag{3}$$

This optimization problem is a Mixed-Integer Program (MIP). The combinatorial nature arises from the cardinality constraint on the support of $d$, which limits the number of active models. A standard way to formulate this is by introducing a binary activation variable $z_m \in \{0, 1\}$ for each model. The full stage-$k$ deployment problem can then be written as:

$$\max_{d, z} \sum_{m \in \mathcal{M}_{\tau_k}} \mu_m^U d_m, \quad \text{s.t.} \quad \sum_m c_m^L d_m \leq b, \sum_m d_m = 1, \ 0 \leq d_m \leq \alpha_m z_m,$$
$$\sum_m z_m = \min(M_{\max}, |\mathcal{M}_{\tau_k}|), \ z_m \in \{0, 1\}.$$

Here, the binary variables $z_m$ explicitly select which models are live for the stage, while the continuous variables $d_m$ represent an optimistic deployment mix. The solution to this MIP $d^*$ maximizes an optimistic performance surrogate using UCBs for rewards ($\mu_m^U$) and LCBs for costs ($c_m^L$), which are derived from data up to $\tau_k$. Specifically, let $\bar{\mu}_m(\tau_k)$ and $\bar{c}_m(\tau_k)$ be the empirical mean reward and cost of model $m$ based on $N_m(\tau_k)$ selections observed up to $\tau_k$. Define the UCBs and LCBs as:

$$\mu_m^U := \mathrm{proj}_{[0,1]}\left(\bar{\mu}_m(\tau_k) + 2f_{rad}(\bar{\mu}_m(\tau_k), N_m(\tau_k) + 1)\right), \tag{4}$$
$$c_m^L := \mathrm{proj}_{[c_1, c_2]}\left(\bar{c}_m(\tau_k) - 2f_{rad}(\bar{c}_m(\tau_k), N_m(\tau_k) + 1)\right), \tag{5}$$

where $\mathrm{proj}_{[a,b]}$ is a projection function onto the interval $[a, b]$ and $f_{rad}(v, n) = \sqrt{\frac{\gamma v}{n}} + \frac{\gamma}{n}$ (for some $\gamma > 0$) is a confidence radius function.

The deployment optimization in Eq. (3) maximizes expected utility subject to the budget $b$, per-model throughput limits $\alpha_m$, and the concurrency cap $|\mathrm{supp}(d)| = \min(M_{\max}, |\mathcal{M}_{\tau_k}|)$. Its solution

$d^*$ determines the active set for stage $k$:

$$\mathcal{D}_k(\mathcal{A}) \leftarrow \{m \in \mathcal{M}_{\tau_k} \mid d_m^* > 0\}. \tag{6}$$

Crucially, the values $d_m^*$ are *not* used as routing probabilities; they serve only to select the most promising feasible models. The deployment set $\mathcal{D}_k(\mathcal{A})$ remains fixed until the next update point $\tau_{k+1}$.

**Request Routing Phase (Intra-Stage).** For each query arriving at time $t \in [\tau_k, \tau_{k+1})$, `StageRoute` performs four steps: (1) *Routing LP.* It solves the linear program (LP) in Eq. (7) to compute the optimal routing distribution $p_t^* = (p_t^*(m))_{m \in \mathcal{D}_k(\mathcal{A})}$ over the *currently deployed models* $\mathcal{D}_k(\mathcal{A})$.

$$\text{RouteOPT:} \quad \max_{p_t \in \Delta(\mathcal{D}_k(\mathcal{A}))} \left\{ \sum_{m \in \mathcal{D}_k(\mathcal{A})} \mu_m^U p_t(m) \,\Big|\, \sum_{m \in \mathcal{D}_k(\mathcal{A})} c_m^L p_t(m) \leq b, \right.$$
$$\left. \sum_{m \in \mathcal{D}_k(\mathcal{A})} p_t(m) = 1, 0 \leq p_t(m) \leq \alpha_m \text{ for } m \in \mathcal{D}_k(\mathcal{A}) \right\}. \tag{7}$$

This LP maximizes the expected reward by combining the current UCBs for reward ($\mu_m^U$) and LCBs for cost ($c_m^L$) while enforcing the per-query budget $b$ (through the cost bounds) and each model's throughput limit $\alpha_m$. (2) *Model selection.* Sample $m_t \sim p_t^*$ and serve the query. (3) *Feedback.* `StageRoute` observes the realized reward $r_t$ and cost $c_t$. (4) *Update.* Refresh $\bar{\mu}_{m_t}, \bar{c}_{m_t}, N_{m_t}$ and recompute $\mu_{m_t}^U, c_{m_t}^L$ for subsequent routing and the next deployment decision.

**Algorithmic Innovations.** While `StageRoute` builds on the principle of "optimism in the face of uncertainty", its architecture is tailored to dynamic LLM deployment and departs from standard bandit formulations. First, it imposes a hierarchical decision structure that mirrors operational practice: the deployment decision is a high-stakes, combinatorial choice whose consequences persist for an entire stage, a form of long-term commitment absent from standard, per-round bandits. Second, staged updates create a structured delay in acting on feedback. Information about non-deployed models cannot influence decisions until the next stage, inducing an exploration-exploitation trade-off that requires anticipating performance over the whole stage, not just the next round. Finally, `StageRoute` decouples deployment from routing execution: `DeployOPT` determines only the active set $\mathcal{D}_k(\mathcal{A})$, while the per-query policy is recomputed online via `RouteOPT`. This separation enables rapid query-level adaptation even when the underlying infrastructure remains fixed during a stage.

## 4 THEORETICAL RESULTS

We analyze `StageRoute` (Algorithm 1) by deriving an upper bound on its cumulative regret and a matching lower bound that applies to any online algorithm for this problem. Together, these results show that `StageRoute` is near-optimal in the worst case.

### 4.1 UPPER BOUND

**Theorem 1.** *Consider* `StageRoute` *running for $T$ queries divided into $K$ stages, with a concurrency cap $M_{\max}$ and $N = |\mathcal{M}_T|$ total models arriving over time. Set the confidence parameter to $\gamma = \Theta(\log(NT/\delta))$ to obtain overall confidence $1 - \delta$. Then the expected regret is bounded by:*

$$\text{Regret}(\texttt{StageRoute}) \leq \mathcal{O}\left( \sqrt{M_{\max} K T \log(NT/\delta)} + \frac{NT}{M_{\max} K} \right).$$

*Choosing $K = \Theta(T^{1/3})$ and $M_{\max} = \Omega(N^{2/3})$ yields* $\text{Regret}(\texttt{StageRoute}) \leq \widetilde{\mathcal{O}}\left( N^{1/3} T^{2/3} \right)$.

The two terms in the bound reflect complementary sources of difficulty. The first term, $\widetilde{\mathcal{O}}\left(\sqrt{M_{\max} K T}\right)$, is the statistical learning cost of routing within the deployed set. The second term, $\widetilde{\mathcal{O}}\left(NT/(M_{\max} K)\right)$, is a structural *model-discovery bottleneck*: when only $M_{\max}$ models can be active at a time across $K$ stages, exploration is throttled. Strong late-arriving models can be missed unless sufficient deployment slots and update frequency are provisioned. Balancing these two terms gives the near-optimal $\widetilde{\mathcal{O}}(T^{2/3})$ rate, which matches the lower bound in Theorem 2.

**Practical guidance.** The bound yields an actionable rule: to approach the optimal rate, set the number of stages to approximately $K \approx T^{1/3}$ (implying a stage length of $T^{2/3}$) and provision a concurrency cap $M_{\max}$ large enough to track new arrivals (ideally $M_{\max} \approx N^{2/3}$ when feasible). However, updating more frequently ($K \gg T^{1/3}$) is also ill-advised. Even if it does not increase regret, it cannot offer further asymptotic improvement due to the lower bound, while needlessly incurring significant computational and operational overhead with each additional deployment stage. This underscores that *exploration capacity*, defined by the concurrency cap and update frequency, is itself a scarce system resource to be optimized. Relying on only a few "top" models is provably suboptimal in a dynamic model pool.

**Why existing analyses do not apply.** Classical MAB and BwK typically assume per-round choices from a static set and lack a hard *concurrency cap*; CMAB selects superarms from a fixed base set and does not model stage-level commitment; streaming bandits allow arrivals but do not couple stage-committed deployment with budget and per-model throughput constraints. Our setting is distinct due to: (i) a concurrency cap that explicitly constrains the support of the deployment optimization, (ii) stage-committed deployment that induces a structured delay in acting on new information, and (iii) the simultaneous enforcement of a long-term budget and per-model throughput limits.

**Proof ideas and new technical elements.** Our proof introduces a *virtual optimal* deployment set to bridge the offline benchmark and the online policy, yielding a clean regret decomposition into stage-level deployment regret and intra-stage routing regret. The routing term is handled with standard confidence arguments. The deployment term requires two new ingredients: (a) a quantification of the *model-discovery bottleneck* caused by the limited concurrency cap and discrete updates (showing how the $NT/(M_{\max}K)$ term arises), and (b) a support-aware sensitivity analysis of the deployment LP (via its dual), bounding how UCB/LCB estimation errors perturb the optimal active set under the concurrency constraint. Together, these yield Theorem 1. These elements differ fundamentally from standard learning-regret analyses and may inform further work on staged, combinatorial online decision problems. Complete details appear in Appendix C.

## 4.2 LOWER BOUND

**Theorem 2.** *For any online policy $\mathcal{A}$ and any choice of update frequency $K$ and concurrency cap $M_{\max}$, there exists a stochastic, piecewise-stationary LLM-routing instance such that the expected regret against the time-varying oracle satisfies*

$$\text{Regret}(\mathcal{A}) \geq \Omega(T^{2/3}).$$

This $\Omega(T^{2/3})$ bound captures the intrinsic difficulty of continually tracking the best model as capabilities evolve. The construction mirrors real LLM ecosystems: in each *batch* a (newer) model is marginally stronger than the rest, and the identity of the strongest model changes over batches. Importantly, the *baseline level* of rewards also drifts upward over time, reflecting that even "weaker" new releases can outperform old ones. This is unlike classic lower bounds that keep suboptimal arms at a fixed mean (e.g., $1/2$) and change only the identity of the best arm.

**Proof ideas and Intuition.** We construct a family of "streaming" instances that mimics a live LLM marketplace: the time horizon is split into approximately $T^{1/3}$ epochs, in each batch a different model is slightly better than the others, and the whole performance frontier drifts upward across batches (so even "weaker" newcomers can surpass yesterday's best). We choose the batch length and performance gaps so that any algorithm cannot reliably identify it with the limited information available before the epoch ends. Because the identity of the best model changes next batch, information gained earlier quickly goes stale. Any policy is thus forced into repeated "partial discovery", incurring a nontrivial loss in each batch, and summing over all batches yields total regret on the order of $T^{2/3}$. Full details are given in Appendix D.

**New technical elements.** Two aspects differ from standard MAB/CMAB lower bounds: (i) we do not use a fixed baseline where only the identity of the best arm flips. Here the *entire* frontier drifts, matching LLM practice; and (ii) the hardness persists even if the system can redeploy every round and keep all models live, so the rate is intrinsic to tracking an evolving frontier, not an artifact of staging or capacity limits.

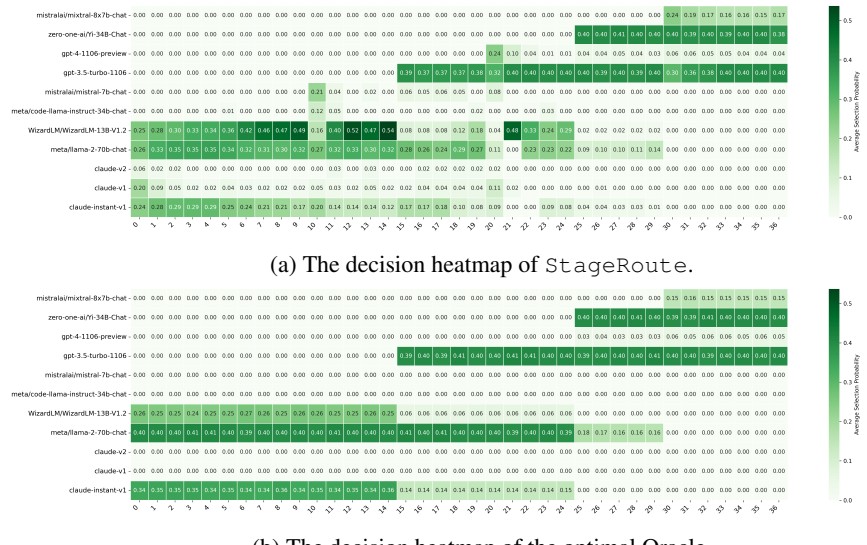

(a) The decision heatmap of `StageRoute`.

(b) The decision heatmap of the optimal Oracle.

Figure 2: Comparison of decision heatmaps for `StageRoute` and the Oracle with $M_{\max} = 5, b = 0.001$, update interval=1000. Darker colors indicate higher selection probabilities.

**Discussion.** Theorem 2 conveys a practical message: even with aggressive adaptivity and large live sets, there is a fundamental rate limit on how quickly a system can keep up as the performance frontier shifts. Our upper bound matches this lower bound up to logarithmic factors with respect to the number of queries $T$, establishing near-optimality.

## 5 EXPERIMENTS

**Datasets and Candidate LLMs.** We evaluate on RouterBench (Hu et al., 2024), covering 36,497 queries across eight datasets in English and Chinese (commonsense, knowledge, dialogue, math, code, and RAG). Each query includes responses from 11 LLMs with per-query scores and costs. Full dataset descriptions and the model list appear in Appendix E.

**Baselines.** We compare `StageRoute` against three baselines. The first is an *oracle* that, with full knowledge of all performance and cost statistics, always selects the optimal deployment set, serving as an upper bound on achievable performance. The second is a *greedy* strategy that, at each update point, selects the $M_{\max}$ models with the highest utility, computed as the ratio between the UCB of performance and the LCB of cost. This approach can be viewed as a variant of a UCB algorithm where selection is based on the UCB of the utility metric. The third baseline is a *uniform sampling* strategy, which randomly selects models for deployment and may substantially exceed the budget. To our knowledge, no existing methods are specifically designed for this LLM deployment problem.

**Implementation Details.** We simulate a total of $T = 36,497$ rounds. In each round, a query is sampled uniformly at random from the dataset. The algorithm then selects a model to serve the query and subsequently receives the performance score and associated cost. Initially, 5 models are available. Thereafter, for every 5,000 queries, a new model becomes eligible for deployment, following the release-date ordering in Table 3. We set the confidence parameter $\gamma = 0.1$. All reported results are averaged over 10 independent runs. The experiments involve solving mixed-integer programming (MIP) subproblems using the Gurobi Optimizer (v12.0.1, academic license) on a machine equipped with a 12th Gen Intel(R) Core(TM) i9-12900HX processor.

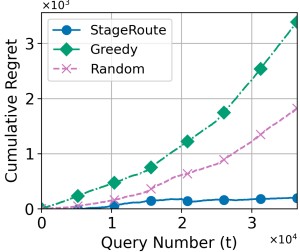

Figure 3: Cumulative regret.

**Computational Overhead.** Our two-stage design keeps the computational overhead low, allowing the entire experimental run to complete in under 10 minutes. The deployment MIP is solved only

infrequently on small instances (taking sub-seconds), while the per-query routing involves a tiny LP over the active set that executes in milliseconds. Therefore, using either Gurobi or an open-source solver (e.g., CBC, GLPK, HiGHS) is feasible. Furthermore, since parameters change only slightly between iterations, we can leverage warm-starting to further accelerate computation. For the routing LP, we can reuse the previous basis or solution; similarly, for the deployment MIP, we can provide the prior active set as an initial feasible solution (MIP start), given the minimal changes to the candidate pool (Zhang et al., 2025a).

**Overall Performance.** We first present the main results under a representative setting ($M_{\max} = 5, b = 0.001$, update interval=1000), and then conduct a detailed sensitivity analysis across key hyperparameters. Figure 3 shows the cumulative regret under this default configuration. Across all settings we test (see Figure 5 for a full overview), our algorithm exhibits consistently slow regret growth, substantially outperforming the baselines. Notably, while the uniform sampling strategy appears to outperform the greedy baseline in some cases, this is an artifact of its tendency to significantly overspend the budget, an issue we will analyze further in the performance-cost evolution.

**Optimal Model Set Identification.** As our work emphasizes the importance of deployment, we first analyze whether `StageRoute` can identify the optimal model set. Figure 2 compares the decision probabilities of `StageRoute` and the Oracle under the representative setting mentioned above. The horizontal axis represents deployment intervals, while the vertical axis (bottom to top) corresponds to the model arrival order. It is evident that when a new model arrives, `StageRoute` initially explores it before quickly converging to the new optimal model set, closely mirroring the Oracle's behavior. This confirms that our deployment strategy is effective at tracking the optimal available model set.

**Performance-Cost Evolution.** To further validate our algorithm's efficiency, Figure 4 illustrates the performance-cost trajectory for each algorithm, again under the same representative setting. Colors transition from blue (initial stages) to red (final stages). The figure shows that, except during initial exploration and periods when new models arrive, the operating points of our algorithm closely track those of the Oracle. In contrast, the greedy strategy proves overly conservative, while uniform sampling consistently violates the budget for suboptimal performance. These observations reinforce our central claim: selecting a high-quality set of models for deployment is fundamental to achieving efficient routing, and `StageRoute` successfully balances high performance with strict budget adherence.

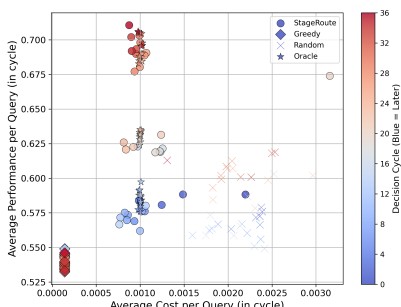

Figure 4: Performance-cost evolution of different algorithms.

**Sensitivity Analysis.** We now analyze `StageRoute`'s sensitivity to key hyperparameters.

(1) *Impact of $M_{\max}$.* Figures 3, 5a, and 5b present results for different $M_{\max}$ values under a budget of $b = 0.001$ and an update interval of 1000. The results demonstrate that `StageRoute` adapts well to this parameter, maintaining robust performance across all settings.

(2) *Effect of Deployment Update Interval.* Figures 3, 5c, and 5d illustrate the impact of varying the deployment update interval with $M_{\max} = 5$ and $b = 0.001$. An interval of 1000 rounds yields the lowest regret, highlighting the importance of selecting an appropriate update frequency.

(3) *Effect of Budget Constraint.* Figures 3 and 5e compare performance under different budget constraints. Counterintuitively, a more relaxed budget leads to higher regret. This phenomenon can be attributed to two factors. First, a larger budget also raises the performance of the Oracle, making the benchmark more challenging. Second, we use a fixed confidence radius $\gamma$ for all settings; in practice, increasing $\gamma$ in proportion to the budget may be beneficial.

**Extending to State-of-the-Art Models.** To verify that our `StageRoute` framework applies to the latest, most powerful models, we conduct additional simulations incorporating recent LLMs. These results, detailed in Appendix E, confirm that `StageRoute` continues to achieve minimal regret.

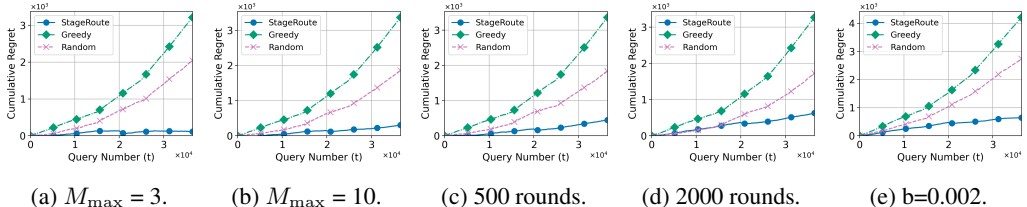

Figure 5: Cumulative regret under varying hyperparameters. The default setting is $M_{\max} = 5$, update interval = 1000 rounds, and $b = 0.001$ (Figure 3).

## 6 CONCLUSION

In this paper, we introduced `StageRoute`, a novel framework for online LLM deployment and routing. We are the first to formalize this problem in a dynamic setting with streaming LLM model arrivals, addressing the challenge of selecting an optimal deployment set under a strict concurrency cap. Our algorithm manages deployment at discrete stages while tactically routing queries in real time, respecting both budget and per-model throughput limits. We established the near-optimality of our algorithm with theoretical analysis, including matching upper and lower bounds, and demonstrated its practical effectiveness through extensive experiments on real-world benchmarks.

## ACKNOWLEDGEMENTS

This work was supported in part by the National Science Foundation (NSF) grants 2148309, 2315614 and 2337914, and the National Institutes of Health (NIH) grant 1R01HL184139-01, and was supported in part by funds from OUSD R&E, NIST, and industry partners as specified in the Resilient & Intelligent NextG Systems (RINGS) program. Any opinions, findings, and conclusions or recommendations expressed in this material are those of the authors and do not necessarily reflect the views of the funding agencies.

## ETHICS STATEMENT

This research focuses on the operational efficiency of LLM systems. By making deployment and routing more cost-effective, our work can broaden access to AI technologies and reduce energy consumption. However, we acknowledge that increased accessibility may also lower the barrier for malicious use of LLMs. Our framework does not mitigate the inherent risks of language models, such as bias or misinformation generation, and should be implemented alongside robust safety and content moderation protocols.

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

# A  RELATED WORK

## A.1  LLM ROUTING

The central aim of LLM routing is to strike the best balance between task performance (e.g., response quality or accuracy) and operational metrics such as cost and latency (Ding et al., 2024; Aggarwal et al., 2024). Existing work follows three main architectural patterns. Ensemble strategies query multiple models in parallel to boost robustness, at the expense of higher cost and latency Wang et al. (2023a); Jiang et al. (2023). Cascade strategies issue queries sequentially—typically starting with a cheaper model and escalating only when necessary—thereby reducing cost but sometimes increasing latency (Chen et al., 2024a; Gupta et al., 2024; Yue et al., 2024; Aggarwal et al., 2024).Direct-routing strategies train a policy or classifier that selects a single LLM per query (Ong et al., 2025; Feng et al., 2025; Zhang et al., 2025b; Zhuang et al., 2025). Benchmark suites such as RouterBench (Hu et al., 2024) facilitate systematic comparison of these approaches. Related work on mixture-of-experts (MoE) models explores routing within a single large model (Du et al., 2022; Fedus et al., 2022; Riquelme et al., 2021). More recently, bandit formulations have been applied to static LLM routing (Wang et al., 2025; Dai et al., 2024; Li, 2025; Nguyen et al., 2024; Poon et al., 2025). Most prior studies, however, assume a fixed set of available models and focus solely on per-query decisions. In contrast, our work model a dynamic model pool with streaming arrivals and introduce staged deployment updates, where the active model set is subject to online selection under a strict concurrency cap, cost budget, and throughput limits. To our knowledge, we are the first to formalize and solve this more realistic and challenging problem.

## A.2  MULTI-ARMED BANDITS

Our formulation builds on the multi-armed-bandit (MAB) paradigm, where an agent maximizes its cumulative payoff through exploration and exploitation in online environment (Auer et al., 2002; Slivkins, 2019). Three MAB extensions are especially pertinent: (i) *Bandits with knapsacks (BwK)*. Here each arm pull yields a reward and consumes limited resources from one or more budgets; the objective is to maximize total reward without overspending (Badanidiyuru et al., 2013; Agrawal & Devanur, 2014; Immorlica et al., 2019; Kesselheim & Singla, 2020; Bernasconi et al., 2024; Guo & Liu, 2025). Our long-term cost constraint fits naturally into this framework. (2) *Streaming bandits.* In this setting new arms arrive over time—often under memory or attention limits—so the agent must adapt to a continually expanding action set (Assadi & Wang, 2020; Jin et al., 2021; Agarwal et al., 2022; Wang, 2023; Li et al., 2023; Shao & Fang, 2025; Zhu & Huang, 2025). The steady appearance of new LLMs places our problem squarely in this category. (3) *Combinatorial Multi-Armed Bandits (CMAB).* The algorithm faces a fixed, known set of base arms from which superarms (subsets) are chosen in each round (Cesa-Bianchi & Lugosi, 2009; Chen et al., 2013; Qin et al., 2014; Li et al., 2016; Chen et al., 2018; Liu et al., 2024; 2025).

The core distinctions arise from the two-level structure and unique constraints inherent to the realistic online LLM deployment and routing problem. Standard models like BwK and streaming bandits lack the combinatorial selection. While CMAB addresses superarm selection, it is fundamentally misaligned with our problem's dynamics: it assumes a static set of base arms and makes per-round decisions, whereas our core challenges are a dynamic model pool and staged, irreversible commitment, where a deployed set remains fixed for a long duration. The regret is thus a function not only of the chosen set but also of the tactical routing policy executed over thousands of queries within that stage. This stateful, hierarchical structure is beyond the scope of traditional bandit formulations. Due to these fundamental differences, existing algorithms and regret analyses are not applicable. Our work bridges this gap by developing a new framework and novel analytical tools tailored to the unique challenges of online LLM deployment and routing problem.

# B    SUMMARY OF NOTATION

Table 2: Table of Notation Used in the System Model and Algorithms

| Symbol | Description |
|---|---|
| *General Parameters & Indices* | |
| $T; t \in [T]$ | Total queries (horizon $T$); $t$ is query index in $[T] = \{1, \ldots, T\}$. |
| $K; k \in [K]$ | Total deployment stages ($K$); $k$ is stage index. |
| $\tau_k$ | Start time step of stage $k$; $k$-th deployment update point, $\tau_k = (k-1)T/K + 1$. |
| $\tau_0$ | Initial time for the algorithm, typically 0. |
| $\mathcal{A}$ | The online deployment and routing algorithm. |
| *LLMs: Availability & Deployment* | |
| $m; t_m$ | An individual LLM $m$; and its availability time $t_m$. |
| $\mathcal{M}_t, \mathcal{M}_{\tau_k}$ | Set of LLMs available at time $t$, and specifically at start of stage $k$. |
| $M_{\max}$ | Maximum number of LLMs that can be simultaneously deployed. |
| $d = (d_m)$ | Deployment decision variable vector in `DeployOPT` over $\mathcal{M}_{\tau_k}$. |
| $d^* = (d_m^*)$ | Optimal deployment decision vector from `DeployOPT` (Eq. 3) at $\tau_k$. |
| $\mathcal{D}_0(\mathcal{A})$ | Initial set of deployed LLMs by algorithm $\mathcal{A}$ (typically empty). |
| $\mathcal{D}_k(\mathcal{A})$ | Set of LLMs deployed by $\mathcal{A}$ in stage $k$ (derived from $d_m^* > 0$). |
| *LLM Performance & Operational Costs* | |
| $\nu_m(\cdot), \mu_m$ | Performance distribution for LLM $m$ (on $[0,1]$) and its mean $\mu_m = \mathbb{E}_{x \sim \nu_m(\cdot)}[x]$. |
| $r_t$ | Realized performance from model $m_t$ for query $t$, $r_t \sim \nu_{m_t}(\cdot)$. |
| $\bar{\mu}_m(\tau_k), \mu_m^U$ | Empirical mean performance (from $N_m(\tau_k)$ obs. up to $\tau_k$) and UCB for $\mu_m$. |
| $p_{\text{in}}, p_{\text{out}}$ | Per-token prices for input and output. |
| $c_{m_t}^{(\text{in})}, c_{m_t}^{(\text{out})}; c_{m_t}$ | Input cost, output cost; and total cost $c_{m_t} = c_{m_t}^{(\text{in})} + c_{m_t}^{(\text{out})}$ for $m_t$ on query $t$. |
| $\xi_m(\cdot)$ | True (unknown) distribution of output token length for LLM $m$. |
| $\mathbb{E}[c_m]$ | True expected operational cost of LLM $m$. |
| $\bar{c}_m(\tau_k)$ | Empirical mean operational cost of LLM $m$ based on $N_m(\tau_k)$ selections up to $\tau_k$. |
| $c_m^L$ | Lower Confidence Bound (LCB) on the expected operational cost $\mathbb{E}[c_m]$. |
| $c_1, c_2$ | Fixed lower and upper bounds for any $c_{m_t}, 0 < c_1 \leq c_2 < \infty$. |
| *Routing & Constraints* | |
| $m_t$ | LLM selected by the algorithm to handle query $t$. |
| $p_t(m)$ | Probability assigned by a routing policy to LLM $m \in \mathcal{D}_k(\mathcal{A})$ for query $t$. |
| $p_t^* = (p_t^*(m))$ | Optimal routing probabilities from `RouteOPT` (Eq. 7). |
| $b$ | Long-term average operational cost budget per query. |
| $\alpha_m$ | Throughput limit (maximum load share / selection probability constraint) for LLM $m$. |
| $\Delta(\mathcal{S})$ | Set of all probability distributions over a set of LLMs $\mathcal{S}$. |
| *Parameter Estimation & Confidence Bounds* | |
| $N_m(\tau_k)$ | Number of times LLM $m$ has been selected and observed up to $\tau_k$. |
| $f_{rad}(v, n), \gamma$ | Confidence radius function $f_{rad}(v, n) = \sqrt{\gamma v/n} + \gamma/n$ (with parameter $\gamma > 0$). |
| $\text{proj}_{[a,b]}(x)$ | Projection of value $x$ onto the interval $[a, b]$. |
| *Offline Benchmark & Regret* | |
| $V(b, \mathcal{S})$ | Optimal Performance Rate Function: max expected performance from set $\mathcal{S}$. |
| $\text{supp}(p)$ | Support of a probability distribution $p$. |
| $\text{OPT}^*$ | Expected cumulative reward of the time-varying offline optimal policy. |
| $\text{Regret}(\mathcal{A})$ | Regret of online algorithm $\mathcal{A}$. |

## C  TECHNICAL ANALYSIS

In this section, we analyze the regret of `StageRoute` (Algorithm 1). Recall that in our setting, let $N = |\mathcal{M}_T|$ denote the total number of models that may arrive over the course of the time horizon. We assume that $N$ is significantly larger than $M_{\max}$, the maximum number of models that can be deployed simultaneously. Moreover, the number of update points $K$ is assumed to be much smaller than the total number of queries $T$. These assumptions reflect practical constraints: it is typically infeasible to deploy all available models—including those released in the future—due to resource limitations, and continuously updating the deployed LLM pool in real time is operationally impractical.

### C.1  CONCENTRATION INEQUALITY

We employ the following standard concentration inequality and related supporting lemmas.

**Lemma 1** (Kleinberg et al. (2008); Babaioff et al. (2015)). *Consider a sequence of random variables $x_1, x_2, \ldots, x_n$. Let $\bar{x} = \frac{1}{n}\sum_{i=1}^n x_i$ be the empirical average and $v = \frac{1}{n}\sum_{i=1}^n \mathbb{E}[x_i|x_1, \ldots, x_{i-1}]$ (if $i = 1$, the expectation is unconditional). If the values $x_i$ are in $[0, 1]$ (e.g., performance $r_t$, or cost $c_t$ under our assumption), then for each $\gamma > 0$,*

$$\mathbb{P}[|v - \bar{x}| \leq f_{rad}(\bar{x}, n) \text{ and } f_{rad}(\bar{x}, n) \leq 3f_{rad}(v, n)] \geq 1 - \exp(-\Omega(\gamma)), \tag{8}$$

*where $f_{rad}(v, n) = \sqrt{\frac{\gamma v}{n}} + \frac{\gamma}{n}$. This result also holds if $x_1, \ldots, x_n$ are independent samples from a distribution with mean $v$ and values in $[0, 1]$.*

For clarity and to simplify the application of concentration inequalities, we assume throughout this analysis that all operational costs $c_{m_t}$ are bounded such that $0 < c_1 \leq c_{m_t} \leq c_2 \leq 1$. This ensures that costs, like rewards (which are in $[0, 1]$), fall within a $[0, 1]$ range (or a sub-interval thereof). This assumption does not affect the order of the regret bounds, as any scaling factors related to a broader cost range would typically be absorbed into the constants within the $\mathcal{O}(\cdot)$ notation.

**Lemma 2** (Babaioff et al. (2015), adapted). *Let $\mathcal{D}_k(\mathcal{A})$ be the set of deployed models in stage $k$. For any vectors $\boldsymbol{a} = (a_m)_{m \in \mathcal{D}_k(\mathcal{A})}$ and $\boldsymbol{n} = (n_m)_{m \in \mathcal{D}_k(\mathcal{A})}$ where $a_m, n_m \geq 0$,*

$$\sum_{m \in \mathcal{D}_k(\mathcal{A})} f_{rad}(a_m, n_m)n_m \leq \sqrt{\gamma M_k \left(\sum_{m \in \mathcal{D}_k(\mathcal{A})} a_m n_m\right)} + \gamma M_k.$$

*where $M_k = |\mathcal{D}_k(\mathcal{A})|$.*

**Lemma 3** (Babaioff et al. (2015), adapted). *Let $\hat{\mu}_m(t) = (\sum_{s < t : m_s = m} r_s)/(N_m(t) + 1)$ be the empirical average performance and $\hat{c}_m(t) = (\sum_{s < t : m_s = m} c_s)/(N_m(t) + 1)$ be the empirical average cost for model $m \in \mathcal{D}_k(\mathcal{A})$ based on $N_m(t)$ plays before time $t$ within the current stage $k$. Then, for every $m \in \mathcal{D}_k(\mathcal{A})$ and time $t \in [\tau_k, \tau_{k+1})$, with probability $1 - e^{-\Omega(\gamma)}$ (i.e., on the event $\mathcal{E}$):*

$$|\hat{\mu}_m(t) - \mu_m| \leq 2f_{rad}(\hat{\mu}_m(t), N_m(t) + 1) \tag{9}$$

$$|\hat{c}_m(t) - \mathbb{E}[c_m]| \leq 2f_{rad}(\hat{c}_m(t), N_m(t) + 1) \tag{10}$$

*Proof.* Follows from applying Lemma 1 to the sequence of observed performances $r_s$ (for $m_s = m$) and observed costs $c_s$ (for $m_s = m$). For a fixed model $m$, the rewards $r_s$ (when $m_s = m$) are i.i.d. samples from $\nu_m(\cdot)$ with mean $\mu_m$. Similarly, costs $c_s$ (when $m_s = m$) are effectively i.i.d. samples with mean $\mathbb{E}[c_m]$. Thus, the conditional expectation $\mathbb{E}[x_i|x_1, \ldots, x_{i-1}]$ in Lemma 1 becomes the true mean $\mu_m$ (or $\mathbb{E}[c_m]$). The derivation is analogous to Lemma 4.3 of Babaioff et al. (2015). For instance, for performance:

$$|\hat{\mu}_m(t) - \mu_m| = \left|\frac{\sum_{s < t : m_s = m} r_s}{N_m(t) + 1} - \frac{(N_m(t) + 1)\mu_m}{N_m(t) + 1}\right|$$

$$\leq \frac{N_m(t)}{N_m(t) + 1} f_{rad}(\hat{\mu}_m(t), N_m(t)) + \frac{\mu_m}{N_m(t) + 1} \quad \text{(from Lemma 1 structure)}$$

$$\leq f_{rad}(\hat{\mu}_m(t), N_m(t) + 1) + \frac{\mu_m}{N_m(t) + 1}$$

$$\leq 2f_{rad}(\hat{\mu}_m(t), N_m(t) + 1)$$

The argument for cost is similar due to the assumption $c_m \in [c_1, c_2] \subseteq [0, 1]$.  □

C.2 REGRET DECOMPOSITION WITH TIME-VARYING BENCHMARK

To analyze the regret of StageRoute (Algorithm 1) over the horizon $T$, we decompose the total regret into components corresponding to the model deployment and request routing phases.

**Definition 1** (Optimal Performance within Deployed Set). *For a given stage $k$ (time interval $[\tau_k, \tau_{k+1})$ of length $T_k = \tau_{k+1} - \tau_k$) where StageRoute (during its Model Deployment Phase) deploys the set $\mathcal{D}_k = \mathcal{D}_k(\mathcal{A}) \subseteq \mathcal{M}_{\tau_k}$, $V(b, \mathcal{D}_k)$ represents the optimal expected reward per query achievable using only models from the deployed set $\mathcal{D}_k$. The total optimal expected performance within this stage using $\mathcal{D}_k$ is $OPT_k = T_k \cdot V(b, \mathcal{D}_k)$. Note that $\mathcal{D}_k$ is determined by StageRoute based on $\mathcal{M}_{\tau_k}$ and estimates available at $\tau_k$. Thus, $\mathcal{D}_k$ and consequently $V(b, \mathcal{D}_k)$ (and $OPT_k$) are random variables, dependent on the algorithm's choices and observations up to time $\tau_k$.*

**Definition 2** (Algorithm Performance). *Let ALGO be the total expected reward accumulated by the algorithm over the horizon $T$:*

$$ALGO = \sum_{t=1}^{T} r_t = \sum_{k=1}^{K} \sum_{t=\tau_k}^{\tau_{k+1}-1} r_t.$$

*Let $ALGO_k = \sum_{t=\tau_k}^{\tau_{k+1}-1} r_t$ be the reward accumulated by the algorithm during stage $k$.*

**Lemma 4** (Regret Decomposition with Time-Varying Benchmark). *The total expected regret $\mathcal{R}(T) = OPT^* - \mathbb{E}[ALGO]$ of StageRoute, compared against the optimal time-varying benchmark $OPT^* = \sum_{k=1}^{K} \sum_{t=\tau_k}^{\tau_{k+1}-1} V(b, \mathcal{M}_{\tau_k})$, can be decomposed as:*

$$\mathcal{R}(T) = \underbrace{\mathbb{E}\left[\sum_{k=1}^{K} \sum_{t=\tau_k}^{\tau_{k+1}-1} (V(b, \mathcal{D}_k) - r_t)\right]}_{\mathcal{R}_{\text{routing}}(T)} + \underbrace{\mathbb{E}\left[\sum_{k=1}^{K} \sum_{t=\tau_k}^{\tau_{k+1}-1} (V(b, \mathcal{M}_{\tau_k}) - V(b, \mathcal{D}_k))\right]}_{\mathcal{R}_{\text{deploy}}(T)}$$

*where:*

- *$\mathcal{R}_{\text{routing}}(T)$ is the total expected routing regret, accumulating the per-query difference between the optimal expected performance with the deployed set $V(b, \mathcal{D}_k)$ and the realized reward $r_t$, summed over all queries and stages.*

- *$\mathcal{R}_{\text{deploy}}(T)$ is the total expected deployment regret, accumulating the per-query difference in optimal expected performance achievable with the full set of available models $V(b, \mathcal{M}_{\tau_k})$ versus the deployed set $V(b, \mathcal{D}_k)$, summed over all queries and stages.*

*Proof.* We start with the definition of the total expected regret:

$$\mathcal{R}(T) = OPT^* - \mathbb{E}[ALGO].$$

Using the definition $OPT^* = \sum_{k=1}^{K} \sum_{t=\tau_k}^{\tau_{k+1}-1} V(b, \mathcal{M}_{\tau_k})$ and $ALGO = \sum_{k=1}^{K} \sum_{t=\tau_k}^{\tau_{k+1}-1} r_t$:

$$\mathcal{R}(T) = \sum_{k=1}^{K} \sum_{t=\tau_k}^{\tau_{k+1}-1} V(b, \mathcal{M}_{\tau_k}) - \mathbb{E}\left[\sum_{k=1}^{K} \sum_{t=\tau_k}^{\tau_{k+1}-1} r_t\right].$$

Note that $\mathcal{M}_{\tau_k}$ (the set of models available at time $\tau_k$) depends on the fixed model arrival times $t_m$ and the stage start time $\tau_k$. According to the system model (Section 2), $\tau_k = (k-1)T/K + 1$ and the stage length $T_k = (\tau_{k+1} - \tau_k) = T/K$ are deterministic. Consequently, the set $\mathcal{M}_{\tau_k}$ and the benchmark value $V(b, \mathcal{M}_{\tau_k})$ are deterministic for each stage $k$. The randomness in the regret decomposition arises from the algorithm's choices, specifically the selection of $\mathcal{D}_k$ (which determines $V(b, \mathcal{D}_k)$) and the subsequent routing decisions leading to the realized rewards $r_t$.

Since $V(b, \mathcal{M}_{\tau_k})$ is deterministic for each $k$ and constant for $t \in [\tau_k, \tau_{k+1} - 1)$, the sum $\sum_{k=1}^{K} \sum_{t=\tau_k}^{\tau_{k+1}-1} V(b, \mathcal{M}_{\tau_k})$ is also deterministic. Thus, it can be written as

$\mathbb{E}\left[\sum_{k=1}^{K}\sum_{t=\tau_k}^{\tau_{k+1}-1} V(b, \mathcal{M}_{\tau_k})\right]$. We add and subtract the term $\mathbb{E}\left[\sum_{k=1}^{K}\sum_{t=\tau_k}^{\tau_{k+1}-1} V(b, \mathcal{D}_k)\right]$:

$$\mathcal{R}(T) = \mathbb{E}\left[\sum_{k=1}^{K}\sum_{t=\tau_k}^{\tau_{k+1}-1} V(b, \mathcal{M}_{\tau_k})\right] - \mathbb{E}\left[\sum_{k=1}^{K}\sum_{t=\tau_k}^{\tau_{k+1}-1} r_t\right]$$

$$= \mathbb{E}\left[\sum_{k=1}^{K}\sum_{t=\tau_k}^{\tau_{k+1}-1} V(b, \mathcal{M}_{\tau_k})\right] - \mathbb{E}\left[\sum_{k=1}^{K}\sum_{t=\tau_k}^{\tau_{k+1}-1} V(b, \mathcal{D}_k)\right]$$

$$+ \mathbb{E}\left[\sum_{k=1}^{K}\sum_{t=\tau_k}^{\tau_{k+1}-1} V(b, \mathcal{D}_k)\right] - \mathbb{E}\left[\sum_{k=1}^{K}\sum_{t=\tau_k}^{\tau_{k+1}-1} r_t\right].$$

Now, we combine terms using the linearity of expectation:

$$\mathcal{R}(T) = \mathbb{E}\left[\sum_{k=1}^{K}\sum_{t=\tau_k}^{\tau_{k+1}-1} \left(V(b, \mathcal{M}_{\tau_k}) - V(b, \mathcal{D}_k)\right)\right]$$

$$+ \mathbb{E}\left[\sum_{k=1}^{K}\sum_{t=\tau_k}^{\tau_{k+1}-1} \left(V(b, \mathcal{D}_k) - r_t\right)\right].$$

This expression matches the claimed decomposition, identifying the deployment regret $\mathcal{R}_{\text{deploy}}(T)$ and the routing regret $\mathcal{R}_{\text{routing}}(T)$ as defined in the lemma statement. Alternatively, letting $\text{OPT}_k = T_k V(b, \mathcal{D}_k) = \sum_{t=\tau_k}^{\tau_{k+1}-1} V(b, \mathcal{D}_k)$ and $\text{ALGO}_k = \sum_{t=\tau_k}^{\tau_{k+1}-1} r_t$, the routing regret can be written as $\mathbb{E}[\sum_{k=1}^{K}(\text{OPT}_k - \text{ALGO}_k)]$. Similarly, the deployment regret can be written as $\mathbb{E}[\sum_{k=1}^{K} T_k(V(b, \mathcal{M}_{\tau_k}) - V(b, \mathcal{D}_k))]$. $\qquad\square$

This decomposition provides an accurate picture of the algorithm's performance. The deployment regret $\mathcal{R}_{\text{deploy}}(T)$ isolates the loss incurred specifically by `StageRoute`'s potentially suboptimal selection $\mathcal{D}_k$ (during its Model Deployment Phase) from the available set $\mathcal{M}_{\tau_k}$, measured against the best possible rate $V(b, \mathcal{M}_{\tau_k})$ achievable with those available models. Bounding $\mathcal{R}_{\text{deploy}}(T)$ involves analyzing how effectively the Model Deployment Phase of `StageRoute` identifies the optimal subset of size at most $M_{\max}$ from $\mathcal{M}_{\tau_k}$ based on its estimates. The routing regret $\mathcal{R}_{\text{routing}}(T)$ remains the sum of per-query differences between the optimal expected performance using the deployed models $\mathcal{D}_k$ and the actual realized rewards $r_t$. Lemma 11 (or subsequent analysis) addresses the term $\mathbb{E}[\sum_{t=\tau_k}^{\tau_{k+1}-1}(V(b, \mathcal{D}_k) - r_t) \mid \mathcal{D}_k]$ which contributes to $\mathcal{R}_{\text{routing}}(T)$.

## C.3   ANALYSIS OF DEPLOYMENT REGRET

We now analyze the deployment regret component $\mathcal{R}_{\text{deploy}}(T)$ as defined in Lemma 4:

$$\mathcal{R}_{\text{deploy}}(T) = \mathbb{E}\left[\sum_{k=1}^{K} T_k \left(V(b, \mathcal{M}_{\tau_k}) - V(b, \mathcal{D}_k)\right)\right].$$

This quantity captures the cumulative expected performance loss across all stages, incurred when the `StageRoute` algorithm selects a subset $\mathcal{D}_k$ at stage $k$ based on estimated model statistics at time $\tau_k$, instead of deploying the optimal subset from the full set of available models $\mathcal{M}_{\tau_k}$.

The deployment regret arises from two complementary sources:

1. **Parameter Uncertainty:** Inaccurate estimates of model performance ($\mu_m$) and cost ($\mathbb{E}[c_m]$) may result in suboptimal deployment decisions. This source corresponds to models that have already been deployed in one or more of the previous $k-1$ stages.

2. **Model Discovery Bottleneck:** The constraint that at most $M_{\max}$ models can be deployed concurrently may exclude promising but underexplored models—particularly newly arrived ones—from being included in $\mathcal{D}_k$. This prevents timely evaluation and utilization, contributing to additional regret. This case pertains to models that have not been selected in any of the preceding $k-1$ stages.

These two components are complementary in nature and together constitute the total deployment regret $\mathcal{R}_{\text{deploy}}(T)$. In the following analysis, we will provide regret bounds for each source.

For the analysis, we recall that $V(b, \mathcal{S})$ is the optimal performance rate for a model set $\mathcal{S}$ (defined in Section 2). Let $V_k^* = V(b, \mathcal{M}_{\tau_k})$ be the optimal rate achievable using all models available at the start of stage $k$, and $V_k = V(b, \mathcal{D}_k)$ be the optimal rate achievable using the subset $\mathcal{D}_k \subseteq \mathcal{M}_{\tau_k}$ selected by StageRoute. $V_k^*$ is deterministic given $\tau_k$, while $V_k$ (through $\mathcal{D}_k$) is a random variable. A key assumption for bounding the deployment regret due to parameter uncertainty involves the Lagrange multipliers associated with the budget constraint in the definition of $V_k^*$. Let $\lambda_k^* \geq 0$ be this optimal Lagrange multiplier. We assume that $\lambda_k^*$ is uniformly bounded by a constant $\Lambda$ for all $k$. This is a common assumption in the analysis of learning algorithms with budget constraints and is often justified when standard regularity conditions (such as Slater's condition, which we assume in Assumption 1) hold for the underlying optimization problems, particularly given that our problem parameters (rewards, costs, $\alpha_m$) are bounded and expected costs $\mathbb{E}[c_m]$ are lower-bounded by $c_1 > 0$.

**Confidence Bounds and Good Event.** Let $\tau_k$ be the start of stage $k$. $N_m(\tau_k)$ is the play count for model $m$ before $\tau_k$. Define confidence radii using Lemma 1 and 3 with $\gamma = \Theta(\log(NT/\delta))$:

$$\text{rad}_\mu(m, \tau_k) = 2f_{rad}(\bar{\mu}_m(\tau_k), N_m(\tau_k) + 1)$$
$$\text{rad}_c(m, \tau_k) = 2f_{rad}(\bar{c}_m(\tau_k), N_m(\tau_k) + 1) \quad \text{(Since costs } c_m \in [c_1, c_2] \subseteq [0, 1])$$

Let $\mathcal{E}_k$ be the good event at time $\tau_k$ where, for all $m \in \mathcal{M}_{\tau_k}$, the confidence bounds based on $\gamma$ hold:

$$\mu_m \leq \mu_m^U(\tau_k) \quad \text{and} \quad \mu_m^U(\tau_k) \leq \mu_m + \text{rad}_\mu(m, \tau_k)$$
$$\mathbb{E}[c_m] - \text{rad}_c(m, \tau_k) \leq c_m^L(\tau_k) \quad \text{and} \quad c_m^L(\tau_k) \leq \mathbb{E}[c_m]$$

Let $\mathcal{E} = \cap_{k=1}^K \mathcal{E}_k$. By a union bound over all $N$ models in the universe $\mathcal{M}_T$ and $K$ stages, $\mathbb{P}(\mathcal{E}) \geq 1 - \delta$. We condition the analysis on $\mathcal{E}$. (Note: $\mu_m^U$ is $\bar{\mu}_m(\tau_k) + 2f_{rad}(\bar{\mu}_m(\tau_k), N_m(\tau_k) + 1)$ and $c_m^L$ is $\bar{c}_m(\tau_k) - 2f_{rad}(\bar{c}_m(\tau_k), N_m(\tau_k) + 1)$ as per Eq. (6,7) in the algorithm description, projected onto $[0, 1]$ and $[c_1, c_2]$ respectively. The inequalities above capture the desired properties on the good event $\mathcal{E}_k$).

**Bounding the Per-Stage Deployment Gap due to Estimation Uncertainty.**

**Lemma 5** (Per-Stage Deployment Gap Bound). *Let $V_k^* = V(b, \mathcal{M}_{\tau_k})$ be the optimal rate with available models at stage $k$. Let $\mathcal{D}_k = \text{supp}(d^*)$ be the set selected by the Model Deployment Phase of* StageRoute *based on $\mathcal{M}_{\tau_k}$ and estimates at $\tau_k$, via solution $d^*$ (from Eq. equation 3). Let $V_k = V(b, \mathcal{D}_k)$. On the good event $\mathcal{E}_k$, the deployment gap for stage $k$ is bounded as:*

$$V_k^* - V_k \leq \sum_{m \in \mathcal{M}_{\tau_k}} \left( \text{rad}_\mu(m, \tau_k) + \lambda_k^* \text{rad}_c(m, \tau_k) \right) d_m^*$$

*where $\lambda_k^*$ is the optimal dual variable for the budget constraint in the problem defining $V_k^*$, assumed to be $\leq \Lambda$.*

*Proof.* Let $d^{\text{opt},k}$ be an optimal solution achieving $V_k^* = V(b, \mathcal{M}_{\tau_k})$. Let $d^*$ be the solution found by StageRoute's Model Deployment Phase (using Eq. equation 3) at $\tau_k$ when optimizing over $\mathcal{M}_{\tau_k}$ using $\mu_m^U(\tau_k)$ and $c_m^L(\tau_k)$. Let $\mathcal{D}_k = \text{supp}(d^*)$. Let $V_k = V(b, \mathcal{D}_k)$.

On the event $\mathcal{E}_k$, the confidence bounds hold for all $m \in \mathcal{M}_{\tau_k}$. Specifically, $\mu_m \leq \mu_m^U(\tau_k) \leq \mu_m + \text{rad}_\mu(m, \tau_k)$ and $\mathbb{E}[c_m] - \text{rad}_c(m, \tau_k) \leq c_m^L(\tau_k) \leq \mathbb{E}[c_m]$. The true optimal solution $d^{\text{opt},k}$ for the set $\mathcal{M}_{\tau_k}$ satisfies $\sum_{m \in \mathcal{M}_{\tau_k}} \mathbb{E}[c_m] d_m^{\text{opt},k} \leq b$, uses $\leq M_{\max}$ models from $\mathcal{M}_{\tau_k}$, etc. Since $c_m^L(\tau_k) \leq \mathbb{E}[c_m]$ on $\mathcal{E}_k$, we have $\sum_{m \in \mathcal{M}_{\tau_k}} c_m^L(\tau_k) d_m^{\text{opt},k} \leq \sum_{m \in \mathcal{M}_{\tau_k}} \mathbb{E}[c_m] d_m^{\text{opt},k} \leq b$. Thus, $d^{\text{opt},k}$ is a feasible solution for the optimization problem solved by StageRoute (Eq. equation 3 applied to $\mathcal{M}_{\tau_k}$).

By the optimality of $d^*$ for StageRoute's deployment objective over $\mathcal{M}_{\tau_k}$:

$$\sum_{m \in \mathcal{M}_{\tau_k}} \mu_m^U(\tau_k) d_m^* \geq \sum_{m \in \mathcal{M}_{\tau_k}} \mu_m^U(\tau_k) d_m^{\text{opt},k} \tag{11}$$

Using the confidence bounds on $\mathcal{E}_k$ for $m \in \mathcal{M}_{\tau_k}$:

$$\sum_{m \in \mathcal{M}_{\tau_k}} \mu_m^U(\tau_k) d_m^* \leq \sum_{m \in \mathcal{M}_{\tau_k}} (\mu_m + \mathrm{rad}_\mu(m, \tau_k)) d_m^* = \sum_{m \in \mathcal{M}_{\tau_k}} \mu_m d_m^* + \sum_{m \in \mathcal{M}_{\tau_k}} \mathrm{rad}_\mu(m, \tau_k) d_m^*$$

$$\sum_{m \in \mathcal{M}_{\tau_k}} \mu_m^U(\tau_k) d_m^{\mathrm{opt},k} \geq \sum_{m \in \mathcal{M}_{\tau_k}} \mu_m d_m^{\mathrm{opt},k} = V_k^*$$

Substituting these into Eq. equation 11:

$$\sum_{m \in \mathcal{M}_{\tau_k}} \mu_m d_m^* + \sum_{m \in \mathcal{M}_{\tau_k}} \mathrm{rad}_\mu(m, \tau_k) d_m^* \geq V_k^*$$

Rearranging:

$$\sum_{m \in \mathcal{M}_{\tau_k}} \mu_m d_m^* \geq V_k^* - \sum_{m \in \mathcal{M}_{\tau_k}} \mathrm{rad}_\mu(m, \tau_k) d_m^* \tag{12}$$

This bounds the true performance of the distribution $d^*$ chosen by the algorithm. Now we relate this to $V_k = V(b, \mathcal{D}_k)$, the optimal performance within the chosen set $\mathcal{D}_k = \mathrm{supp}(d^*) \subseteq \mathcal{M}_{\tau_k}$. The distribution $d^*$ is supported on $\mathcal{D}_k$ and uses at most $M_{\max}$ models (due to the constraint in Eq. equation 3). We examine its feasibility w.r.t. the true budget constraint. On $\mathcal{E}_k$:

$$\sum_{m \in \mathcal{D}_k} \mathbb{E}[c_m] d_m^* \leq \sum_{m \in \mathcal{D}_k} (c_m^L(\tau_k) + \mathrm{rad}_c(m, \tau_k)) d_m^* \leq b + \sum_{m \in \mathcal{D}_k} \mathrm{rad}_c(m, \tau_k) d_m^*$$

Let $\delta_c(d^*) = \sum_{m \in \mathcal{D}_k} \mathrm{rad}_c(m, \tau_k) d_m^*$. Using sensitivity analysis/duality, relating $V_k = V(b, \mathcal{D}_k)$ to the performance of $d^*$ which is feasible for budget $b + \delta_c(d^*)$:

$$V_k = V(b, \mathcal{D}_k) \geq V(b + \delta_c(d^*), \mathcal{D}_k) - \lambda_k^* \delta_c(d^*)$$

where $\lambda_k^* \leq \Lambda$. Since $d^*$ is feasible for $V(b + \delta_c(d^*), \mathcal{D}_k)$:

$$V(b + \delta_c(d^*), \mathcal{D}_k) \geq \sum_{m \in \mathcal{D}_k} \mu_m d_m^* = \sum_{m \in \mathcal{M}_{\tau_k}} \mu_m d_m^*$$

Combining these:

$$V_k \geq \left( \sum_{m \in \mathcal{M}_{\tau_k}} \mu_m d_m^* \right) - \lambda_k^* \delta_c(d^*)$$

$$\geq \left( V_k^* - \sum_{m \in \mathcal{M}_{\tau_k}} \mathrm{rad}_\mu(m, \tau_k) d_m^* \right) - \lambda_k^* \sum_{m \in \mathcal{D}_k} \mathrm{rad}_c(m, \tau_k) d_m^* \quad \text{(Using Eq. equation 12)}$$

Rearranging gives the result (noting $d_m^* = 0$ for $m \notin \mathcal{D}_k$):

$$V_k^* - V_k \leq \sum_{m \in \mathcal{M}_{\tau_k}} \mathrm{rad}_\mu(m, \tau_k) d_m^* + \lambda_k^* \sum_{m \in \mathcal{D}_k} \mathrm{rad}_c(m, \tau_k) d_m^*$$

Since $d_m^* = 0$ for $m \notin \mathcal{D}_k$, the second sum can also be written over $\mathcal{M}_{\tau_k}$:

$$V_k^* - V_k \leq \sum_{m \in \mathcal{M}_{\tau_k}} \left( \mathrm{rad}_\mu(m, \tau_k) + \lambda_k^* \mathrm{rad}_c(m, \tau_k) \right) d_m^*$$

$\square$

**Cumulative Deployment Regret from Estimation Uncertainty.** Summing the per-stage deployment gaps caused by estimation errors gives the learning component of the deployment regret.

**Lemma 6** (Deployment Regret from Estimation Uncertainty). *Assume the optimal dual variables $\lambda_k^*$ are uniformly bounded by $\Lambda$. Let $K$ be the total number of stages. Set the confidence parameter $\gamma = \Theta(\log(NT/\delta))$, where $N = |\mathcal{M}_T|$. Then the component of total expected deployment regret due to parameter uncertainty, denoted $\mathcal{R}_{\mathrm{deploy,learn}}(T)$, is bounded by:*

$$\mathcal{R}_{\mathrm{deploy,learn}}(T) \leq \mathcal{O}\left( \sqrt{T \log(NT/\delta) \cdot \min(N, KM_{\max})} + M_{\max} K \log(NT/\delta) \right).$$

*Proof.* The total expected deployment regret due to parameter uncertainty is given by

$$\mathcal{R}_{\text{deploy,learn}}(T) = \mathbb{E}\left[\sum_{k=1}^{K} T_k \left(V(b, \mathcal{M}_{\tau_k}) - V(b, \mathcal{D}_k)\right)\right],$$

where $V_k^* = V(b, \mathcal{M}_{\tau_k})$ and $V_k = V(b, \mathcal{D}_k)$. We condition on the good event $\mathcal{E} = \cap_{k=1}^{K} \mathcal{E}_k$, which holds with probability at least $1 - \delta$. On this event, the confidence bounds for $\mu_m$ and $\mathbb{E}[c_m]$ hold for all models $m$ and stages $k$. From Lemma 5, on event $\mathcal{E}_k$:

$$V_k^* - V_k \le \sum_{m \in \mathcal{M}_{\tau_k}} \left(\text{rad}_\mu(m, \tau_k) + \lambda_k^* \text{rad}_c(m, \tau_k)\right) d_{k,m}^*.$$

Let $C_m(\tau_k) = \text{rad}_\mu(m, \tau_k) + \Lambda \text{rad}_c(m, \tau_k)$, using the uniform bound $\lambda_k^* \le \Lambda$. Since $d_{k,m}^* = 0$ for $m \notin \mathcal{D}_k = \text{supp}(d_k^*)$, we have:

$$T_k(V_k^* - V_k) \le T_k \sum_{m \in \mathcal{D}_k} C_m(\tau_k) d_{k,m}^*.$$

The term $d_{k,m}^*$ represents the selection weight for model $m$ in the deployment optimization at stage $k$. Thus, $\sum_{m \in \mathcal{D}_k} C_m(\tau_k) d_{k,m}^*$ is a weighted average of the combined confidence radii for the deployed models. We are considering the case where the model has already been selected in the previous $k - 1$ stages. Thus, we can bound the sum of $T_k(V_k^* - V_k)$ by terms related to $\sum_{k=1}^{K} \sum_{m \in \mathcal{D}_k} n_{k,m} C_m(\tau_k)$. Specifically,

$$\sum_{k=1}^{K} T_k(V_k^* - V_k) \le \mathcal{O}(1) \sum_{k=1}^{K} \sum_{m \in \mathcal{D}_k} n_{k,m} C_m(\tau_k).$$

Let's analyze the sum $S = \sum_{k=1}^{K} \sum_{m \in \mathcal{D}_k} n_{k,m} C_m(\tau_k)$. Recall $C_m(\tau_k) = 2 f_{rad}(\bar{\mu}_m(\tau_k), N_m(\tau_k) + 1) + 2\Lambda f_{rad}(\bar{c}_m(\tau_k), N_m(\tau_k) + 1)$. Since rewards $\bar{\mu}_m(\tau_k) \in [0, 1]$ and costs $\bar{c}_m(\tau_k) \in [c_1, c_2] \subseteq [0, 1]$ (with $c_1 > 0$), we have $f_{rad}(v, n) = \sqrt{\frac{\gamma v}{n}} + \frac{\gamma}{n} \le \sqrt{\frac{\gamma}{n}} + \frac{\gamma}{n}$ for $v \in [0, 1]$. So, $C_m(\tau_k) \le 2(1 + \Lambda) \left(\sqrt{\frac{\gamma}{N_m(\tau_k)+1}} + \frac{\gamma}{N_m(\tau_k)+1}\right)$. Let $C' = 2(1 + \Lambda)$.

$$S \le C' \sum_{k=1}^{K} \sum_{m \in \mathcal{D}_k} n_{k,m} \left(\sqrt{\frac{\gamma}{N_m(\tau_k) + 1}} + \frac{\gamma}{N_m(\tau_k) + 1}\right).$$

We can rewrite the sum by first summing over all models $m \in \mathcal{M}_T$ (the set of all $N$ possible models) and then over the stages $k$ in which $m$ was deployed and played:

$$S \le C' \sum_{m \in \mathcal{M}_T} \sum_{k: m \in \mathcal{D}_k \text{ and } n_{k,m} > 0} n_{k,m} \left(\sqrt{\frac{\gamma}{N_m(\tau_k) + 1}} + \frac{\gamma}{N_m(\tau_k) + 1}\right).$$

For a fixed model $m$, let $N_m(T)$ be the total number of times it is played up to $T$. Let $n_{m,s_j}$ be the number of times $m$ is played during the $j$-th stage (denoted $s_j$) in which it is deployed. Let $N_m(\tau_{s_j})$ be the total number of plays of $m$ before stage $s_j$. The sum for model $m$ is:

$$S_m = \sum_{j=1}^{K_m'} n_{m,s_j} \left(\sqrt{\frac{\gamma}{N_m(\tau_{s_j}) + 1}} + \frac{\gamma}{N_m(\tau_{s_j}) + 1}\right),$$

where $K_m'$ is the number of stages model $m$ is played. We use the standard inequalities for such sums:

- $\sum_{j=1}^{K_m'} n_{m,s_j} \sqrt{\frac{\gamma}{N_m(\tau_{s_j})+1}} \le \sqrt{\gamma} \sum_{i=1}^{N_m(T)} \frac{1}{\sqrt{(\text{plays of } m \text{ before current block})+1}} \le \sqrt{\gamma} \cdot 2\sqrt{N_m(T)}$.

- $\sum_{j=1}^{K_m'} n_{m,s_j} \frac{\gamma}{N_m(\tau_{s_j})+1} \le \gamma \sum_{i=1}^{N_m(T)} \frac{1}{(\text{plays of } m \text{ before current block})+1} \le \gamma \cdot (1 + \ln N_m(T))$.

So, $S_m \leq \mathcal{O}(\sqrt{\gamma N_m(T)} + \gamma \log N_m(T))$. (We absorb constants into $\mathcal{O}(\cdot)$ for now, and will re-introduce $C'$ later). Thus, $S \leq C' \sum_{m \in \mathcal{M}_T \text{ s.t. } N_m(T) > 0} \mathcal{O}(\sqrt{\gamma N_m(T)} + \gamma \log N_m(T))$.

The sum is over models that were actually played. Let $\mathcal{M}_P = \{m \in \mathcal{M}_T \mid N_m(T) > 0\}$. The number of models in $\mathcal{D}_k$ is $|\mathcal{D}_k| \leq M_{\max}$. So, $\sum_{m \in \mathcal{M}_P} \sqrt{N_m(T)} \leq \sqrt{|\mathcal{M}_P| \sum_{m \in \mathcal{M}_P} N_m(T)} = \sqrt{|\mathcal{M}_P| T}$ by Cauchy-Schwarz. Since at most $M_{\max}$ models are deployed in any stage, and there are $K$ stages, the number of distinct models ever deployed is $|\mathcal{M}_P| \leq \min(N, K M_{\max})$.

The sum $\sum_{m \in \mathcal{M}_P} \log N_m(T)$. If $N_m(T) \geq 1$, then $\log N_m(T) \geq 0$. This sum is at most $M_{\max} \log(T/M_{\max})$ if $M_{\max}$ models share $T$ plays, or more generally bounded by $M_{\max} K$ (if each of $M_{\max}$ models gets played at least once in each of $K$ stages, its $\log N_m(T)$ contributes, and $N_m(T)$ could be small). A more careful bound for the sum of log terms: $\sum_{m \in \mathcal{M}_P} \log N_m(T) \leq M_{\max} K \log T$. So, $S \leq C' \left( \mathcal{O}(\sqrt{\gamma T \cdot \min(N, K M_{\max})}) + \mathcal{O}(\gamma M_{\max} K) \right)$. The log terms are typically absorbed into the $\gamma$ term. The expectation $\mathbb{E}[S]$ includes the good event (probability $1 - \delta$) and the bad event (probability $\delta$). On the bad event, the regret in one stage is at most $T_k$, so total $T$. $\mathcal{R}_{\text{deploy,learn}}(T) \leq \mathbb{E}[S] + \delta T$. If $\delta = \mathcal{O}(1/T)$, then $\delta T = \mathcal{O}(1)$. Substituting $C' = 2(1 + \Lambda)$ and $\gamma = \Theta(\log(NT/\delta))$, then we complete the proof. □

**Deployment Regret from Model Discovery Bottleneck.** The constraint of deploying at most $M_{\max}$ models simultaneously, $|\mathcal{D}_k(\mathcal{A})| \leq M_{\max}$, introduces a structural challenge, particularly when new models frequently become available or the total pool of models $\mathcal{M}_T$ is large. This challenge is the model discovery bottleneck: identifying truly superior models among many new, unevaluated candidates can be delayed.

The `StageRoute` algorithm employs UCBs for rewards ($\mu_m^U$) and LCBs for costs ($c_m^L$) in its `DeployOPT` phase (Eq. equation 3). For a model $m$ that is newly available at stage $k$ (i.e., $t_m \leq \tau_k$ and its count of previous selections $N_m(\tau_k) = 0$), its initial empirical averages $\bar{\mu}_m(\tau_k)$ and $\bar{c}_m(\tau_k)$ are set based on priors. The confidence radius $f_{rad}(v, N_m(\tau_k) + 1) = \sqrt{\frac{\gamma v}{N_m(\tau_k)+1}} + \frac{\gamma}{N_m(\tau_k)+1}$ becomes large for $N_m(\tau_k) = 0$. Specifically, with $N_m(\tau_k) + 1 = 1$:

$$\mu_m^U(\tau_k) = \text{proj}_{[0,1]} \left( \bar{\mu}_m(\tau_k) + 2 \left( \sqrt{\gamma \bar{\mu}_m(\tau_k)} + \gamma \right) \right),$$
$$c_m^L(\tau_k) = \text{proj}_{[c_1, c_2]} \left( \bar{c}_m(\tau_k) - 2 \left( \sqrt{\gamma \bar{c}_m(\tau_k)} + \gamma \right) \right).$$

Assuming priors are chosen such that new models are treated optimistically (or if $\gamma$ is sufficiently large), $\mu_m^U(\tau_k)$ will be close to 1 (e.g., if $\bar{\mu}_m(\tau_k) = 0$, $\mu_m^U(\tau_k) = \text{proj}_{[0,1]}(2\gamma) \approx 1$ for appropriate $\gamma$) and $c_m^L(\tau_k)$ will be close to $c_1$ (e.g., if $\bar{c}_m(\tau_k) = c_1$, $c_m^L(\tau_k) = \text{proj}_{[c_1, c_2]}(c_1 - 2(\sqrt{\gamma c_1} + \gamma)) \approx c_1$, noting $c_1, c_2 \in [0, 1]$). Let these optimistic initial values be $U_{init}$ and $L_{init}$ respectively.

Consider an update point $\tau_k$. Let $\mathcal{N}_{new,k} \subseteq \mathcal{M}_{\tau_k}$ be the set of models that are new at or before $\tau_k$ and have not yet been deployed ($N_m(\tau_k) = 0$). All models in $\mathcal{N}_{new,k}$ will have nearly identical, highly optimistic ($\mu_m^U, c_m^L) \approx (U_{init}, L_{init})$ values. If the number of such equally optimistic new models, $|\mathcal{N}_{new,k}|$, plus other potentially optimistic (but previously explored) models, exceeds $M_{\max}$, the `DeployOPT` phase must select only $M_{\max}$ models. If the new models in $\mathcal{N}_{new,k}$ dominate the selection pool due to their optimism, `DeployOPT` will choose $M_{\max}$ models from $\mathcal{N}_{new,k}$ (possibly along with some already explored models). Crucially, if there are more than $M_{\max}$ models within $\mathcal{N}_{new,k}$ (or a larger pool of similarly optimistic candidates) that yield effectively the same objective value for `DeployOPT` (because their $\mu_m^U, c_m^L, \alpha_m$ are similar), the selection among these specific candidates becomes arbitrary (e.g., dependent on tie-breaking rules).

A truly superior new model $m^* \in \mathcal{N}_{new,k}$ might thus be part of a large batch of $N_{batch} > M_{\max}$ new models that all appear equally promising to `DeployOPT`. In this scenario, $m^*$ might not be selected for deployment in stage $k$, deferring its evaluation. This deferral means the system misses the opportunity to benefit from $m^*$'s potentially high true performance $\mu_{m^*}$ for the duration of stage $k$, which is $T_k = T/K$ rounds.

**Lemma 7** (Model Discovery Bottleneck Regret). *Let $M_{\max}$ be the maximum number of concurrently deployed models, $T$ be the total time horizon, and $K$ be the number of stages, with each stage*

*having $T_k = T/K$ rounds. The component of expected deployment regret due to the bottleneck in discovering and evaluating all $N$ models, denoted $\mathcal{R}_{\text{deploy,discovery}}(T)$, is bounded by:*

$$\mathcal{R}_{\text{deploy,discovery}}(T) = \mathcal{O}\left(\frac{N \cdot (T/K)}{M_{\max}}\right) = \mathcal{O}\left(\frac{NT}{M_{\max}K}\right).$$

*Proof.* Each of the $N$ models in the universe $\mathcal{M}_T$ needs to be deployed at least once to gather initial empirical data and move its UCB/LCB estimates away from their initial purely optimistic values. We are interested in the total regret incurred until all $N$ models have had at least one such initial deployment opportunity.

Due to the constraint $|\mathcal{D}_k(\mathcal{A})| \leq M_{\max}$, at most $M_{\max}$ distinct models can be deployed and evaluated in any given stage $k$. If the system prioritizes exploring previously undeployed models (which is encouraged by their optimistic UCB/LCB values), it will take a minimum of $K_{explore} = \lceil N/M_{\max}\rceil$ stages to ensure that every model in $\mathcal{M}_T$ (assuming all become available early enough) has been deployed at least once.

Consider a discovery period spanning these first $K_{explore}$ effective stages. During any given stage $j \in [1, K_{explore}]$ within this period, if the set of $M_{\max}$ models deployed, $\mathcal{D}_j(\mathcal{A})$, does not include some model $m^* \in \mathcal{M}_T$ which is (a) available ($m^* \in \mathcal{M}_{\tau_j}$), (b) truly superior to at least one deployed model $m' \in \mathcal{D}_j(\mathcal{A})$, and (c) $m^*$ has not been deployed yet because it is waiting its turn due to the arbitrary selection among many new, equally optimistic models, then regret is incurred. The per-query regret in such a case can be up to 1 (if $\mu_{m^*} \approx 1$ and $\mu_{m'} \approx 0$).

The total number of queries over these $K_{explore}$ stages is $K_{explore} \cdot T_k = \lceil N/M_{\max}\rceil \cdot (T/K)$. During this cumulative period, the system is effectively cycling through the $N$ models. If the selection process within each stage means that, on average, the deployed set is suboptimal because truly good models are among the $(N - j \cdot M_{\max})$ yet to be tried, the system incurs regret. The term $\mathcal{O}(NT/(M_{\max}K))$ represents the cumulative regret if, for a duration equivalent to $N/M_{\max}$ full stages (each of length $T/K$), the system operates with a deployed set that is, on average, $\mathcal{O}(1)$ worse per query than if all models had already been evaluated. This occurs because the $M_{\max}$ slots are occupied by models chosen optimistically, and a superior model might be consistently deferred if it's part of a large pool of indistinguishably optimistic new models.

More formally, consider the $N$ models. Each requires roughly one exploration slot of duration $T_k$. These $N$ slots are processed in parallel groups of $M_{\max}$. This implies approximately $N/M_{\max}$ stages are spent ensuring all models receive initial evaluation. If, during these $N/M_{\max}$ stages, the average deployed set yields $\mathcal{O}(1)$ less reward per query compared to an optimal deployment (had all models been known), the total regret from this discovery phase is $(N/M_{\max}) \cdot (T/K) \cdot \mathcal{O}(1)$. The expectation $\mathbb{E}[\cdot]$ in $\mathcal{R}_{\text{deploy,discovery}}(T)$ averages over the random tie-breaking in `DeployOPT` when faced with multiple equally optimistic new models, and the stochastic arrival pattern of models. The $\mathcal{O}(\cdot)$ notation absorbs constants related to the maximum possible per-query regret (e.g., 1) and the precise nature of average suboptimality during this discovery period. $\square$

This component accounts for the scenarios where truly good models might be systematically delayed in their initial deployment if they frequently arrive alongside many other new models, leading to arbitrary choices among a large pool of initially indistinguishable (optimistic) candidates, subject to the $M_{\max}$ deployment limit over $K$ stages.

**Total Deployment Regret.** The total deployment regret $\mathcal{R}_{\text{deploy}}(T)$ is the sum of the regret from parameter uncertainty (Lemma 6) and the regret from the model discovery bottleneck (Lemma 7).

**Lemma 8** (Total Deployment Regret Bound). *Let $N$ be the total number of models, $M_{\max}$ the maximum deployed models, $T$ the horizon, and $K$ the number of stages. Let $\gamma = \Theta(\log(NT/\delta))$. The total expected deployment regret $\mathcal{R}_{\text{deploy}}(T)$ is bounded by:*

$$\mathcal{R}_{\text{deploy}}(T) \leq \mathcal{O}\left(\sqrt{T\log(NT/\delta) \cdot \min(N, KM_{\max})} + M_{\max}K\log(NT/\delta) + \frac{NT}{M_{\max}K}\right).$$

*Proof.* The total deployment regret is the sum of the bounds from Lemma 6 and Lemma 7:

$$\mathcal{R}_{\text{deploy}}(T) = \mathcal{R}_{\text{deploy,learn}}(T) + \mathcal{R}_{\text{deploy,discovery}}(T)$$

$$\leq \mathcal{O}\left(\sqrt{T \log(NT/\delta) \cdot \min(N, KM_{\max})} + M_{\max}K \log(NT/\delta)\right) + \mathcal{O}\left(\frac{NT}{M_{\max}K}\right)$$

$$= \mathcal{O}\left(\sqrt{T \log(NT/\delta) \cdot \min(N, KM_{\max})} + M_{\max}K \log(NT/\delta) + \frac{NT}{M_{\max}K}\right).$$

$\square$

The total deployment regret bound in Lemma 8 highlights two distinct challenges in the deployment phase. The terms $\mathcal{O}(\sqrt{T \log(NT/\delta) \cdot \min(N, KM_{\max})})$ and $\mathcal{O}(M_{\max}K \log(NT/\delta))$ capture the cost of learning the parameters of models that are considered for deployment. This cost depends on the horizon $T$, the number of deployment slots $M_{\max}$, the number of stages $K$, and logarithmic factors related to the total number of models $N$ and confidence $\delta$. The term $\mathcal{O}(NT/(M_{\max}K))$ reflects the structural cost imposed by the discovery bottleneck: when the universe of models $N$ is large compared to $M_{\max}$ and the number of adaptation opportunities $K$, there is an inherent regret incurred in sequentially exploring models to identify the best ones. This term can dominate if $N$ is very large or $K$ is small, underscoring the importance of the deployment frequency and capacity in dynamic LLM environments.

## C.4 Bounding the Total Routing Regret

We now bound the total routing regret term $\mathcal{R}_{\text{routing}}(T)$ identified in Lemma 4. This involves summing the per-stage regrets incurred by the Request Routing Phase of `StageRoute` due to using estimated model parameters within the deployed sets $\mathcal{D}_k$. This part of the proof follows a similar structure to the analysis of UCB-style algorithms for the Bandits with Knapsacks problem Agrawal & Devanur (2014).

**Lemma 9** (Performance Bound). *Let $ALGO_k = \sum_{t=\tau_k}^{\tau_{k+1}-1} r_t$ be the total observed performance in stage $k$. With probability at least $1 - (M_k T_k) \exp(-\mathcal{O}(\gamma))$,*

$$\left| \sum_{t=\tau_k}^{\tau_{k+1}-1} \left( r_t - \sum_{m \in \mathcal{D}_k(\mathcal{A})} \mu_m^U(t) p_t^*(m) \right) \right| \leq \mathcal{O}\left( \sqrt{\gamma M_k ALGO_k} + \gamma M_k \right).$$

*Proof.* The proof follows the structure of Lemma 4.4 in Babaioff et al. (2015), adapted to our notation for models $m \in \mathcal{D}_k(\mathcal{A})$ and performance $\mu_m$, UCB $\mu_m^U(t)$, chosen model $m_t$, routing distribution $p_t^*$ (from Eq. equation 7), and summing over $t \in [\tau_k, \tau_{k+1} - 1]$ (length $T_k$).

We use Lemma 1 and Lemma 3 (for performance). High probability bounds analogous to (5) and (6) in the source proof hold for sums over $t \in [\tau_k, \tau_{k+1} - 1]$:

$$\left| \sum_t (r_t - \mu_{m_t}) \right| \leq \mathcal{O}(T_k \cdot f_{rad}(\frac{1}{T_k} \sum_t \mu_{m_t}, T_k))$$

$$\left| \sum_t \left( \sum_{m \in \mathcal{D}_k(\mathcal{A})} \mu_m^U(t) p_t^*(m) - \mu_{m_t}^U(t) \right) \right| \leq \mathcal{O}(T_k \cdot f_{rad}(\frac{1}{T_k} \sum_t \mu_{m_t}^U(t), T_k))$$

And analogous to (7) in the source, using Lemma 3 and Lemma 2:

$$\left| \sum_{t=\tau_k}^{\tau_{k+1}-1} (\mu_{m_t} - \mu_{m_t}^U(t)) \right| \le \sum_{t=\tau_k}^{\tau_{k+1}-1} |\mu_{m_t} - \mu_m^U(t)|$$

$$\le \mathcal{O}\left( \sum_{t=\tau_k}^{\tau_{k+1}-1} f_{rad}(\hat{\mu}_{m_t}(t), N_{m_t}(t) + 1) \right)$$

$$\le \mathcal{O}\left( \sum_{m \in \mathcal{D}_k(\mathcal{A})} \sum_{\text{plays } j \text{ of } m \text{ in stage } k} f_{rad}(\hat{\mu}_m(\text{at play } j), N_m(\text{at play } j) + 1) \right)$$

$$\le \mathcal{O}\left( \sum_{m \in \mathcal{D}_k(\mathcal{A})} \left( \sqrt{\gamma(N_m(\tau_{k+1}) - N_m(\tau_k))\mu_m} + \gamma \right) \right)$$

$$\le \mathcal{O}\left( \sqrt{\gamma M_k \left( \sum_{m \in \mathcal{D}_k(\mathcal{A})} \mu_m(N_m(\tau_{k+1}) - N_m(\tau_k)) \right) + \gamma M_k} \right)$$

$$\le \mathcal{O}\left( \sqrt{\gamma M_k \left( \sum_{t=\tau_k}^{\tau_{k+1}-1} \mu_{m_t} \right) + \gamma M_k} \right)$$

Let $A = \sum_{t=\tau_k}^{\tau_{k+1}-1} \sum_{m \in \mathcal{D}_k(\mathcal{A})} \mu_m^U(t) p_t^*(m)$. Combining these bounds using the triangle inequality on $r_t - \sum_m \mu_m^U(t)p_t^*(m) = (r_t - \mu_{m_t}) + (\mu_{m_t} - \mu_{m_t}^U(t)) + (\mu_{m_t}^U(t) - \sum_m \mu_m^U(t)p_t^*(m))$, similar to the source proof structure, leads to an inequality relating $A$ and $\sum r_t = \text{ALGO}_k$. If $\sum r_t \approx \sum \mu_{m_t}$ and $A \approx \sum \mu_{m_t}^U(t)$, then the difference $|\sum(\mu_{m_t} - \mu_{m_t}^U(t))|$ dominates. This typically leads to $A - \mathcal{O}(\sqrt{\gamma M_k A} + \gamma M_k) \le \text{ALGO}_k$ (assuming $\mu^U$ are UCBs and $A \approx \sum \mu_{m_t}^U(t)$). This implies $\sqrt{A} \le \sqrt{\text{ALGO}_k} + \mathcal{O}(\sqrt{\gamma M_k})$. Substituting this back into the bounds for the difference $|\text{ALGO}_k - A|$ yields the claimed result.

$$\left| \sum_{t=\tau_k}^{\tau_{k+1}-1} r_t - \sum_{t=\tau_k}^{\tau_{k+1}-1} \sum_{m \in \mathcal{D}_k(\mathcal{A})} \mu_m^U(t)p_t^*(m) \right| \le \mathcal{O}(\sqrt{\gamma M_k \text{ALGO}_k} + \gamma M_k).$$

$\square$

**Lemma 10** (Cost Bound). *Let $\sum_{t=\tau_k}^{\tau_{k+1}-1} c_t$ be the total observed cost in stage $k$. Let $B_k = b \cdot T_k$ be the effective expected cost budget for the stage. With probability at least $1 - (M_k T_k) \exp(-\mathcal{O}(\gamma))$ (i.e., on event $\mathcal{E}$),*

$$\left| \sum_{t=\tau_k}^{\tau_{k+1}-1} \left( c_t - \sum_{m \in \mathcal{D}_k(\mathcal{A})} c_m^L(t)p_t^*(m) \right) \right| \le \mathcal{O}\left( \sqrt{\gamma M_k B_k} + \gamma M_k \right).$$

*Proof.* The proof mirrors that of Lemma 9 (and Lemma 4.5 in Babaioff et al. (2015)), replacing performance with cost, $\mu_m$ with $\mathbb{E}[c_m]$, $\mu_m^U(t)$ with $c_m^L(t)$, and $r_t$ with $c_t$. The probability distribution $p_t^*$ is from Eq. equation 7. Key steps involve bounding $|\sum c_t - \sum \mathbb{E}[c_{m_t}]|$, $|\sum(\sum_m c_m^L(t)p_t^*(m)) - \sum c_{m_t}^L(t)|$, and $|\sum \mathbb{E}[c_{m_t}] - \sum c_{m_t}^L(t)|$. The last term is bounded similarly using Lemma 3 and Lemma 2:

$$\left| \sum_{t=\tau_k}^{\tau_{k+1}-1} (\mathbb{E}[c_{m_t}] - c_{m_t}^L(t)) \right| \le \mathcal{O}\left( \sqrt{\gamma M_k \left( \sum_{t=\tau_k}^{\tau_{k+1}-1} \mathbb{E}[c_{m_t}] \right) + \gamma M_k} \right).$$

Let $A' = \sum_{t=\tau_k}^{\tau_{k+1}-1} \mathbb{E}[c_{m_t}]$. The algorithm ensures $\sum_{m \in \mathcal{D}_k(\mathcal{A})} c_m^L(t)p_t^*(m) \le b$ at each step $t$. Summing over the stage gives $\sum_{t=\tau_k}^{\tau_{k+1}-1} \sum_{m \in \mathcal{D}_k(\mathcal{A})} c_m^L(t)p_t^*(m) \le b \cdot T_k = B_k$. Let

$X_c = \sum_{t=\tau_k}^{\tau_{k+1}-1} \sum_{m \in \mathcal{D}_k(\mathcal{A})} c_m^L(t) p_t^*(m)$. Then $\left| \sum_{t=\tau_k}^{\tau_{k+1}-1} c_t - X_c \right| \leq \mathcal{O}(\sqrt{\gamma M_k A'} + \gamma M_k)$ where $A'$ is the sum of true expected costs. On event $\mathcal{E}$, $c_m^L(t) \leq \mathbb{E}[c_m]$, so $X_c \leq A'$. Also, $A' \leq X_c + \mathcal{O}(\sqrt{\gamma M_k A'} + \gamma M_k)$. Since $X_c \leq B_k$, it follows that $A' \leq B_k + \mathcal{O}(\sqrt{\gamma M_k A'} + \gamma M_k)$. This implies $\sqrt{A'} \leq \sqrt{B_k} + \mathcal{O}(\sqrt{\gamma M_k})$. Substituting this back into the concentration bounds for $|\sum c_t - X_c|$ (and noting $A' \approx B_k$ in the error term's leading order) yields the final result. $\qquad \square$

**Lemma 11** (Per-Stage Routing Regret Bound). *Let $OPT_k^{val} = \max_p \{ \sum_{m \in \mathcal{D}_k(\mathcal{A})} \mu_m p(m) \mid \sum_{m \in \mathcal{D}_k(\mathcal{A})} \mathbb{E}[c_m] p(m) \leq b, \sum p(m) = 1, 0 \leq p(m) \leq \alpha_m \}$ be the optimal expected reward rate within stage $k$ using $\mathcal{D}_k(\mathcal{A})$ and true parameters. Let $OPT_k = T_k \cdot OPT_k^{val}$ be the total optimal expected performance in stage $k$. Let $ALGO_k = \sum_{t=\tau_k}^{\tau_{k+1}-1} r_t$. Assume $M_k \gamma \leq \mathcal{O}(B_k)$. Then, on the event $\mathcal{E}$ (implying high probability for the bounds within the stage), the routing regret for stage $k$, conditioned on $\mathcal{D}_k$, is bounded by:*

$$\mathbb{E}[OPT_k - ALGO_k \mid \mathcal{D}_k] \leq \mathcal{O}\left(\sqrt{\gamma M_k OPT_k} + \gamma M_k\right)$$

*(The expectation is conditioned on the choice of $\mathcal{D}_k$, which itself depends on information up to $\tau_k$).*

*Proof.* On event $\mathcal{E}$, the following hold:

1. $\mu_m^U(t) \geq \mu_m$ and $c_m^L(t) \leq \mathbb{E}[c_m]$ for all relevant $m, t$.

2. Lemma 9: $|\sum_t (r_t - \sum_m \mu_m^U(t) p_t^*(m))| \leq \mathcal{O}(\sqrt{\gamma M_k ALGO_k} + \gamma M_k)$.

3. Lemma 10: $|\sum_t (c_t - \sum_m c_m^L(t) p_t^*(m))| \leq \mathcal{O}(\sqrt{\gamma M_k B_k} + \gamma M_k)$. Also, $\sum_t c_t \leq B_k + \mathcal{O}(\sqrt{\gamma M_k B_k} + \gamma M_k)$.

From the algorithm's choice of $p_t^*$ (solving Eq. equation 7) and property (1):

$$\sum_{t=\tau_k}^{\tau_{k+1}-1} \sum_{m \in \mathcal{D}_k(\mathcal{A})} \mu_m^U(t) p_t^*(m) \geq \text{OPT}_k$$

Combining this with property (2):

$$\text{ALGO}_k = \sum_t r_t \geq \sum_t \sum_m \mu_m^U(t) p_t^*(m) - \mathcal{O}(\sqrt{\gamma M_k \text{ALGO}_k} + \gamma M_k)$$

$$\text{ALGO}_k \geq \text{OPT}_k - \mathcal{O}(\sqrt{\gamma M_k \text{ALGO}_k} + \gamma M_k)$$

Rearranging and assuming $\text{ALGO}_k \leq \text{OPT}_k$ (regret is non-negative):

$$\text{OPT}_k - \text{ALGO}_k \leq \mathcal{O}(\sqrt{\gamma M_k \text{ALGO}_k} + \gamma M_k)$$

If $\text{ALGO}_k \leq \text{OPT}_k$, then $\sqrt{\text{ALGO}_k} \leq \sqrt{\text{OPT}_k}$.

$$\text{OPT}_k - \text{ALGO}_k \leq \mathcal{O}(\sqrt{\gamma M_k \text{OPT}_k} + \gamma M_k)$$

Taking expectation conditioned on $\mathcal{D}_k$ (and implicitly on $\mathcal{E}$ for the bounds to hold), the result follows. The assumption $M_k \gamma \leq \mathcal{O}(B_k)$ is used in concentration bounds for costs. Moreover, since the estimation errors for both performance and cost are governed by the same concentration inequalities, the expected budget violation is on the same order as the regret. $\qquad \square$

**Lemma 12** (Total Routing Regret Bound). *Let $K$ be the total number of stages. Let $M_k = |\mathcal{D}_k| \leq M_{\max}$. Let $\delta \in (0,1)$ be the desired overall confidence. Set $\gamma = \Theta(\log(NT/\delta))$. Then the total expected routing regret is bounded by:*

$$\mathcal{R}_{\text{routing}}(T) = \mathbb{E}\left[ \sum_{k=1}^{K} (OPT_k - ALGO_k) \right] \leq \mathcal{O}\left( \sqrt{\gamma M_{\max} KT} + K \gamma M_{\max} \right)$$

*Substituting $\gamma = \Theta(\log(NT/\delta))$, this becomes:*

$$\mathcal{R}_{\text{routing}}(T) \leq \mathcal{O}\left( \sqrt{M_{\max} KT \log(NT/\delta)} + K M_{\max} \log(NT/\delta) \right)$$

*Proof.* From Lemma 4, $\mathcal{R}_{\text{routing}}(T) = \sum_{k=1}^{K} \mathbb{E}[\text{OPT}_k - \text{ALGO}_k]$. Here $\text{OPT}_k = T_k V(b, \mathcal{D}_k)$ and $\text{ALGO}_k$ is the algorithm's reward in stage $k$. The outer expectation $\mathbb{E}[\cdot]$ averages over all randomness, including $\mathcal{D}_k$ and the failure of event $\mathcal{E}$ (which occurs with probability $\leq \delta$). Using law of total expectation: $\mathbb{E}[\text{OPT}_k - \text{ALGO}_k] = \mathbb{E}[\mathbb{E}[\text{OPT}_k - \text{ALGO}_k \mid \mathcal{D}_k]]$.

We apply Lemma 11. Choosing $\gamma = \Theta(\log(NT/\delta))$ ensures that event $\mathcal{E}$ holds with probability at least $1 - \delta$. On $\mathcal{E}$:

$$\mathbb{E}[\text{OPT}_k - \text{ALGO}_k \mid \mathcal{D}_k] \leq \mathcal{O}\left(\sqrt{\gamma M_k \text{OPT}_k} + \gamma M_k\right)$$

Taking expectation over $\mathcal{D}_k$:

$$\mathbb{E}[\text{OPT}_k - \text{ALGO}_k] \leq \mathcal{O}\left(\mathbb{E}\left[\sqrt{\gamma M_k \text{OPT}_k}\right] + \gamma \mathbb{E}[M_k]\right) + \delta \cdot T_k$$

Summing this bound over all $K$ stages:

$$\mathcal{R}_{\text{routing}}(T) \leq \sum_{k=1}^{K} \mathcal{O}\left(\mathbb{E}\left[\sqrt{\gamma M_k \text{OPT}_k}\right] + \gamma \mathbb{E}[M_k]\right) + \delta T$$

Using linearity of expectation, $M_k \leq M_{\max}$, Jensen's inequality ($\mathbb{E}[\sqrt{X}] \leq \sqrt{\mathbb{E}[X]}$), and $\mathbb{E}[M_k] \leq M_{\max}$:

$$\mathcal{R}_{\text{routing}}(T) \leq \mathcal{O}\left(\sum_{k=1}^{K} \mathbb{E}\left[\sqrt{\gamma M_{\max} \text{OPT}_k}\right] + \sum_{k=1}^{K} \gamma M_{\max}\right) + \delta T$$

$$\leq \mathcal{O}\left(\sqrt{\gamma M_{\max}} \sum_{k=1}^{K} \sqrt{\mathbb{E}[\text{OPT}_k]} + K \gamma M_{\max}\right) + \delta T$$

Applying Cauchy-Schwarz: $(\sum_{k=1}^{K} \sqrt{\mathbb{E}[\text{OPT}_k]})^2 \leq K \sum_{k=1}^{K} \mathbb{E}[\text{OPT}_k]$. Thus, $\sum_{k=1}^{K} \sqrt{\mathbb{E}[\text{OPT}_k]} \leq \sqrt{K \sum_{k=1}^{K} \mathbb{E}[\text{OPT}_k]}$. Since rewards are in $[0, 1]$, $V(b, \mathcal{D}_k) \leq 1$, so $\text{OPT}_k = T_k V(b, \mathcal{D}_k) \leq T_k$. Summing over $k$: $\sum_{k=1}^{K} \text{OPT}_k \leq \sum_{k=1}^{K} T_k = T$. So, $\sum_{k=1}^{K} \mathbb{E}[\text{OPT}_k] \leq T$. Substituting this upper bound:

$$\mathcal{R}_{\text{routing}}(T) \leq \mathcal{O}\left(\sqrt{\gamma M_{\max} K T} + K \gamma M_{\max}\right) + \delta T$$

If $\delta$ is chosen small enough, the $\delta T$ term is absorbed. This establishes the first form of the bound. Substituting $\gamma = \Theta(\log(NT/\delta))$ yields the second form. $\square$

## C.5 TOTAL REGRET BOUND

The overall regret of `StageRoute` accounts for several sources of suboptimality. The total expected regret $\mathcal{R}(T)$ can now be understood as the sum of two main components:

1. $\mathcal{R}_{\text{routing}}(T)$: The routing regret within deployed sets (Lemma 12).
2. $\mathcal{R}_{\text{deploy}}(T)$: The total deployment regret, encompassing both learning uncertainty and model discovery bottleneck (Lemma 8).

Summing these bounds:

$$\mathcal{R}(T) = \mathcal{R}_{\text{routing}}(T) + \mathcal{R}_{\text{deploy}}(T)$$
$$\leq \mathcal{O}\left(\sqrt{M_{\max} K T \log(NT/\delta)} + K M_{\max} \log(NT/\delta)\right)$$
$$+ \mathcal{O}\left(\sqrt{T \log(NT/\delta) \cdot \min(N, K M_{\max})} + M_{\max} K \log(NT/\delta) + \frac{NT}{M_{\max} K}\right)$$

Combining terms, and noting that for $K \geq 1$, $\sqrt{M_{\max} K T \log(NT/\delta)}$ dominates or is equivalent to $\sqrt{T \log(NT/\delta) \cdot \min(N, K M_{\max})}$ and $K M_{\max} \log(NT/\delta)$:

$$\mathcal{R}(T) \leq \mathcal{O}\left(\sqrt{M_{\max} K T \log(NT/\delta)} + \frac{NT}{M_{\max} K}\right).$$

# D  LOWER BOUND FOR ONLINE LLM ROUTING

We establish a lower bound on the regret for any online LLM routing policy. The construction considers a scenario where the set of competitive models and their performances can evolve over time, divided into batches. Within each batch, the algorithm faces a sequential decision problem of selecting the best among $M$ available models, where the identity of the best model is unknown and can change from batch to batch. To isolate the learning challenge, we make several simplifications. We assume each model invocation incurs a unit cost ($c_{m_t} = 1$), rendering the budget constraint trivial if the per-query budget $b \geq 1$. We also assume that the system can always deploy any of the $M$ models under consideration in a given batch ($M_{\max} \geq M$) and that there are no per-model capacity limits ($\alpha_m = 1$ for all $m$). The core difficulty then lies in continuously learning and adapting to the best-performing model(s) in each batch.

The proof strategy is to construct a class of adversarial problem instances. We will demonstrate that any online algorithm $\mathcal{A}$ must incur significant regret on at least one instance within this class. This argument adapts a batch-based structure common in analyses of learning problems with non-stationary environments.

**Step 1: Construction of Hard Problem Instances.**  Let $\epsilon > 0$ be a small parameter, which will be determined later. The total time horizon of $T$ queries is partitioned into $N_B$ contiguous batches, denoted $\mathcal{B}_1, \ldots, \mathcal{B}_{N_B}$. Each batch $j$ consists of $\Delta = T/N_B$ queries. For simplicity, we assume $T$ is an integer multiple of $N_B$.

For each batch $j \in \{1, \ldots, N_B\}$, we consider a set of $M$ models available to the algorithm $\mathcal{A}$. These models are indexed $i \in \{1, \ldots, M\}$ specifically for the current batch $j$. The performance characteristics of these $M$ models are defined as follows: one model is designated as the "strong" model, and the remaining $M-1$ models are "weak". Let $s_j \in \{1, \ldots, M\}$ be the index of this strong model in batch $j$, chosen uniformly at random by the adversary and unknown to the algorithm. The reward $r_t$ obtained from selecting a model at query $t \in \mathcal{B}_j$ is drawn from a Bernoulli distribution:

- If model $m_{s_j}$ (the strong model for batch $j$, with index $s_j$) is chosen, $r_t \sim \text{Bernoulli}(\mu_H^{(j)})$, where the mean reward is $\mu_H^{(j)} = \frac{1}{2} + j\epsilon$.

- If any other model $m_i$ (where $i \in \{1, \ldots, M\}$ and $i \neq s_j$) from the set of $M$ models for batch $j$ is chosen, $r_t \sim \text{Bernoulli}(\mu_L^{(j)})$, where the mean reward is $\mu_L^{(j)} = \frac{1}{2} + (j-1)\epsilon$.

Rewards from different queries are assumed to be independent. The crucial gap in expected reward between the strong model and any weak model in batch $j$ is $\mu_H^{(j)} - \mu_L^{(j)} = \epsilon$. To ensure that all mean rewards $\mu_H^{(j)}$ and $\mu_L^{(j)}$ lie comfortably within the interval $[0, 1]$, we impose the condition $N_B \epsilon \leq \frac{1}{4}$. This ensures $\mu_H^{(N_B)} = \frac{1}{2} + N_B \epsilon \leq \frac{1}{2} + \frac{1}{4} = \frac{3}{4} < 1$, and the smallest mean, $\mu_L^{(1)} = \frac{1}{2} + (1-1)\epsilon = \frac{1}{2}$, is also valid.

A complete problem instance is characterized by a sequence of strong model indices $(s_1, s_2, \ldots, s_{N_B})$. The actual underlying LLMs corresponding to these indices could differ from batch to batch, but in each batch $j$, the algorithm $\mathcal{A}$ faces a choice among $M$ options with the specified reward structure, and $s_j$ is unknown.

**Step 2: Per-Batch Regret from Identification Difficulty.**  Fix a batch $j \in \{1, \ldots, N_B\}$, and let $s_j^*$ denote the true index of the strong model, unknown to algorithm $\mathcal{A}$. To identify $m_{s_j^*}$ (mean $\mu_H^{(j)}$) from the $M - 1$ weak models (mean $\mu_L^{(j)}$, gap $\epsilon$) with a constant probability $p_{succ} < 1$ (e.g., $p_{succ} = 3/4$), any algorithm requires a query complexity of $\Omega(M\epsilon^{-2})$ Agarwal et al. (2017); Li et al. (2023). This implies there is a universal constant $c_S > 0$ such that at least $c_S M \epsilon^{-2}$ queries are necessary to achieve success probability $p_{succ}$.

We set the batch length $\Delta = (c_S/2)M\epsilon^{-2}$. Since $c_S/2 < c_S$, this choice of $\Delta$ is insufficient for reliable identification with probability $p_{succ}$. Consequently, algorithm $\mathcal{A}$ fails to identify $m_{s_j^*}$ within $\Delta$ queries with at least a constant probability $p_{fail} \geq 1 - p_{succ}$ (e.g., setting $p_{succ} = 3/4$ gives $p_{fail} \geq 1/4$). Let $\mathcal{E}_{fail}$ denote this event of identification failure.

Let $N_{j,weak}$ be the number of queries to weak models in batch $j$. Conditional on $\mathcal{E}_{fail}$, the algorithm lacks knowledge of $s_j^*$. In such a state of confusion (especially when $M \geq 2$), it is expected to select weak models a significant fraction of the time. For instance, if choices upon failure are made nearly uniformly among the $M$ models, weak models are selected $\frac{M-1}{M}\Delta$ times in expectation. Since $M \geq 2$, $(M-1)/M \geq 1/2$. Thus, we can state that $\mathbb{E}[N_{j,weak}|\mathcal{E}_{fail}] \geq c_F\Delta$ for a chosen constant $c_F \geq 1/2$. By the law of total expectation, and since $\mathbb{E}[N_{j,weak}|\mathcal{E}_{fail}^c] \geq 0$:

$$\mathbb{E}[N_{j,weak}] = \mathbb{P}(\mathcal{E}_{fail})\mathbb{E}[N_{j,weak}|\mathcal{E}_{fail}] + \mathbb{P}(\mathcal{E}_{fail}^c)\mathbb{E}[N_{j,weak}|\mathcal{E}_{fail}^c] \geq p_{fail} \cdot c_F\Delta.$$

Using $p_{fail} \geq 1/4$ and $c_F \geq 1/2$, this gives $\mathbb{E}[N_{j,weak}] \geq \frac{1}{4} \cdot \frac{1}{2}\Delta = \frac{\Delta}{8}$. The expected regret in batch $j$ (for a fixed $s_j^*$, averaged over $\mathcal{A}$'s randomness) is $R_j(s_j^*) = \epsilon\mathbb{E}[N_{j,weak}] \geq \epsilon(\Delta/8)$. Since this lower bound does not depend on the specific index $s_j^*$, and the adversary chooses $s_j^*$ uniformly, the expected regret in batch $j$ (averaged over $s_j^*$ and $\mathcal{A}$) is $\mathbb{E}_{\tilde{s}_j}[R_j] \geq \frac{1}{8}\epsilon\Delta$.

**Step 3: Regret Along the Horizon and Parameter Choice.** The expected regret in each batch $j$ is $\mathbb{E}_{\tilde{s}_j}[R_j] \geq \frac{1}{8}\epsilon\Delta$. The total expected regret over $N_B$ batches for algorithm $\mathcal{A}$ is $R_T(\mathcal{A}) = \sum_{j=1}^{N_B} \mathbb{E}_{\tilde{s}_j}[R_j] \geq N_B\frac{1}{8}\epsilon\Delta$. Since $N_B = T/\Delta$, we have $R_T(\mathcal{A}) \geq (T/\Delta)\frac{1}{8}\epsilon\Delta = \frac{1}{8}T\epsilon$.

We have set the batch length $\Delta = (c_S/2)M\epsilon^{-2}$ in Step 2. The other primary constraint, from Step 1, is $N_B\epsilon \leq 1/4$, which implies $(T/\Delta)\epsilon \leq 1/4$. Substituting $\Delta = (c_S/2)M\epsilon^{-2}$ into this constraint:

$$\frac{T}{(c_S/2)M\epsilon^{-2}}\epsilon \leq \frac{1}{4} \implies \frac{2T\epsilon^3}{c_S M} \leq \frac{1}{4} \implies \epsilon^3 \leq \frac{c_S M}{8T}.$$

To maximize the lower bound on regret $\frac{1}{8}T\epsilon$, we choose $\epsilon$ to be as large as possible, subject to this constraint. We set $\epsilon = \left(\frac{c_S M}{8T}\right)^{1/3}$. This choice ensures $\epsilon^3 = \frac{c_S M}{8T}$.

Substituting this $\epsilon$ into the total regret expression:

$$R_T(\mathcal{A}) \geq \frac{1}{8}T\left(\frac{c_S M}{8T}\right)^{1/3} = \frac{1}{8}T\frac{c_S^{1/3}M^{1/3}}{8^{1/3}T^{1/3}} = \frac{1}{8}\frac{c_S^{1/3}}{2}M^{1/3}T^{2/3} = \frac{c_S^{1/3}}{16}M^{1/3}T^{2/3}.$$

Since $c_S > 0$ is a universal constant from the identification complexity, let $C = c_S^{1/3}/16$. Then $C > 0$. Thus, $R_T(\mathcal{A}) \geq CM^{1/3}T^{2/3}$.

Finally, we verify the conditions on our parameter choices:

1. Value of $\Delta$: $\epsilon^2 = \left(\frac{c_S M}{8T}\right)^{2/3}$. $\Delta = (c_S/2)M\epsilon^{-2} = (c_S/2)M\left(\frac{c_S M}{8T}\right)^{-2/3} = (c_S/2)M\frac{(8T)^{2/3}}{(c_S M)^{2/3}} = (c_S/2)M\frac{8^{2/3}T^{2/3}}{c_S^{2/3}M^{2/3}} = (c_S/2)M\frac{4T^{2/3}}{c_S^{2/3}M^{2/3}} = \frac{c_S}{2}\frac{4M^{1/3}T^{2/3}}{c_S^{2/3}} = 2c_S^{1/3}M^{1/3}T^{2/3}$. Let $c_\Delta = 2c_S^{1/3}$. So $\Delta = c_\Delta M^{1/3}T^{2/3}$. We need $1 \leq \Delta \leq T$ for $\Delta$ to be a valid batch length. $\Delta \leq T \implies c_\Delta M^{1/3}T^{2/3} \leq T \implies M^{1/3} \leq T^{1/3}/c_\Delta \implies T \geq (c_\Delta M^{1/3})^3 = c_\Delta^3 M$. This implies $T$ must be sufficiently large relative to $M$. $\Delta \geq 1$ generally holds for large $T$ if $M \geq 1$.

2. Constraint $N_B\epsilon \leq 1/4$: This constraint was used to determine the choice of $\epsilon$. With $\epsilon^3 = \frac{c_S M}{8T}$:
$$\frac{T\epsilon^3}{(c_S/2)M} = \frac{T\left(\frac{c_S M}{8T}\right)}{(c_S/2)M} = \frac{c_S M/8}{c_S M/2} = \frac{1/8}{1/2} = \frac{1}{4}.$$
Thus, $(T/\Delta)\epsilon = N_B\epsilon = 1/4$ is satisfied by this construction.

All conditions are met for suitable choices of $T$ relative to $M$, given the universal constant $c_S > 0$ and our choices for $p_{fail}$ (e.g., $1/4$) and $c_F$ (e.g., $1/2$) which determine the factor $1/8$ in the per-batch regret. Thus, for any online routing algorithm $\mathcal{A}$, there exists a problem instance in our constructed class for which its expected regret is bounded below by $R_T(\mathcal{A}) \geq CM^{1/3}T^{2/3}$ for some constant $C > 0$ (specifically $C = c_S^{1/3}/16$). For a fixed number of models $M \geq 2$. In this case, the lower bound becomes $\Omega(T^{2/3})$, matching the statement in Theorem 2.

This completes the proof.

# E    EXPERIMENTAL DETAILS

This appendix provides additional details on the experimental setup, benchmarks, and results presented in Section 5.

## E.1    REAL-WORLD BENCHMARK: ROUTERBENCH

**Dataset Details.** Our primary evaluation is based on the RouterBench dataset (Hu et al., 2024), a comprehensive benchmark with 36,497 queries sampled from eight diverse NLP datasets. These datasets cover both Chinese and English and span a broad spectrum of tasks: commonsense reasoning (HellaSwag (Zellers et al., 2019), WinoGrande (Sakaguchi et al., 2021), ARC Challenge (Clark et al., 2018)), knowledge-based understanding (MMLU (Hendrycks et al., 2021)), open-domain dialogue (MT-Bench (Zheng et al., 2023)), mathematical reasoning (GSM8K (Cobbe et al., 2021)), code generation (MBPP (Austin et al., 2021)), and retrieval-augmented generation (RAG).

**Candidate LLMs and Metrics.** For each query, RouterBench provides pre-computed responses from the 11 LLMs listed in Table 3. The table shows the models ordered by their release date, along with their average performance score and average cost per query across the entire dataset.

Table 3: List of LLMs sorted by release date, along with their per-query average performance score and average cost.

| Model | Avg Performance | Avg Cost |
|---|---|---|
| claude-instant-v1 | 0.5900 | $0.001236 |
| claude-v1 | 0.6480 | $0.005870 |
| claude-v2 | 0.5116 | $0.006153 |
| meta/llama-2-70b-chat | 0.6059 | $0.001337 |
| WizardLM/WizardLM-13B-V1.2 | 0.5392 | $0.000142 |
| meta/code-llama-instruct-34b-chat | 0.5040 | $0.000550 |
| mistralai/mistral-7b-chat | 0.4999 | $0.000139 |
| gpt-3.5-turbo-1106 | 0.6867 | $0.000709 |
| gpt-4-1106-preview | 0.8048 | $0.007943 |
| zero-one-ai/Yi-34B-Chat | 0.7153 | $0.000558 |
| mistralai/mixtral-8x7b-chat | 0.6504 | $0.000414 |

## E.2    ADDITIONAL SIMULATION RESULTS

To validate our framework's applicability to the rapidly evolving frontier of SOTA models, we constructed a synthetic benchmark using 15 recent, high-performance LLMs.

**Candidate LLMs and Metrics.** We consider a set of 15 LLMs, summarized in Table 4, ordered by their official release dates. For each model, we report both performance metrics Chiang et al. (2024) and associated costs. No single metric can capture the full complexity of LLM performance. We chose Elo / area scores as they are community standards (e.g., Chatbot Arena), ensuring transparency and reproducibility. Importantly, our framework is metric-agnostic. The performance signal can be replaced by any other quantifiable performance measure, such as task-specific accuracy, user satisfaction scores, or a composite utility function combining multiple objectives.

Note that the naming of AI models can be highly nuanced, and multiple versions may exist under a similar label. For example, for Gemini 2.5 Pro, we used data corresponding to the `Gemini-2.5-Pro-Preview-05-06` version. Additionally, performance scores may fluctuate over time due to model updates, shifts in the user voting population, or changes in evaluation benchmarks. Similarly, costs may vary across time or across versions. The data reported here corresponds to the specific versions and conditions used in our experiments.

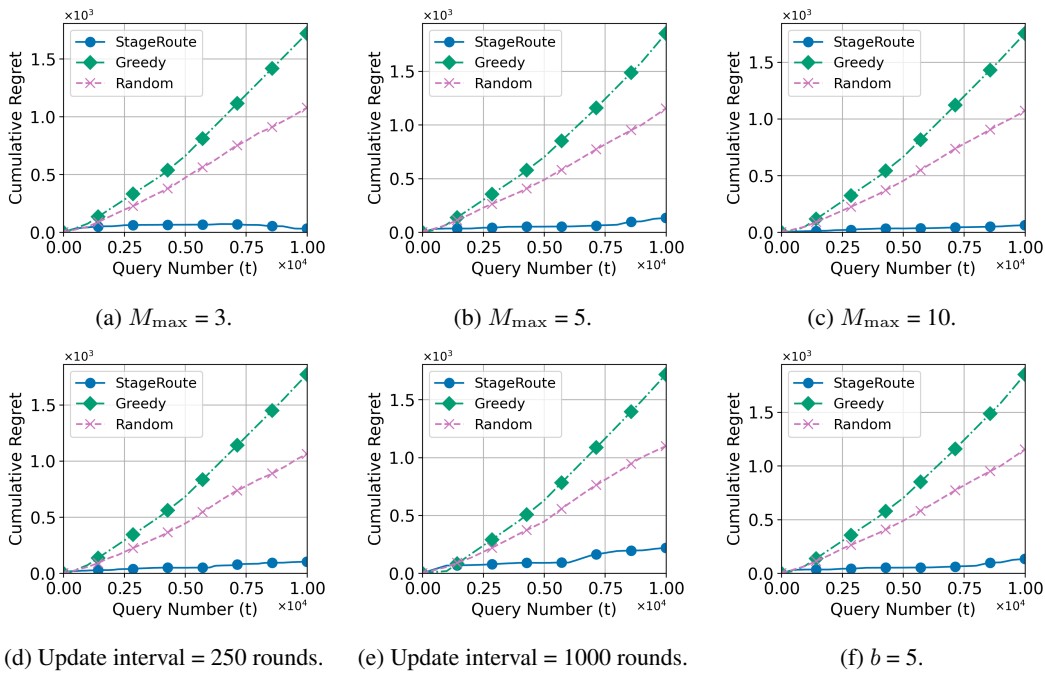

Figure 6: Regret with varying parameters.

**Simulation Setup.** These 15 models are introduced sequentially, with the first 5 available at time $t = 1$ and one additional model becoming available every 1,000 rounds until all are accessible. Each model's performance is normalized to lie within the $[0, 1]$ interval. For each model $m$, we set its budget weight parameter $\alpha_m = 0.4$, except for Yi-Lightning, for which we set $\alpha_m = 1$. This is because Yi-Lightning is the cheapest model initially available to the algorithm, ensuring the feasibility of the mixed-integer optimization problem in every round. At each round, if the algorithm selects a model, it receives a noisy performance and cost observation, where the noise is sampled from a Gaussian distribution centered around the true value. In any real-world application, LLM performance metrics are inherently bounded. Our simulation implicitly respects the $[0, 1]$ range by clipping noisy rewards to this interval, with the noise variance chosen such that the probability of generating values outside the range is negligible.

**Sensitivity Analysis.** Figure 6 presents the sensitivity analysis for `StageRoute` on the synthetic benchmark. The results are consistent with our findings on RouterBench, demonstrating the robustness of our algorithm across different model suites and settings. `StageRoute` consistently achieves low regret, adapting effectively to the concurrency cap, update interval, and budget.

Table 4: LLM comparison by release date: performance (Arena Score/Elo Chiang et al. (2024)) and output cost (per 1M tokens).

| Model | Performance | Cost |
| --- | --- | --- |
| GPT-4o | 1336 | $10.00 |
| Gemini-1.5-pro-002 | 1302 | $5.00 |
| Yi-Lightning | 1287 | $0.14 |
| o1-mini | 1304 | $0.60 |
| Llama 3.3 70B Instruct | 1257 | $0.40 |
| DeepSeek-R1 | 1358 | $0.55 |
| Gemini 2.0 Flash | 1380 | $1.50 |
| Claude 3.7 Sonnet | 1300 | $15.00 |
| Hunyuan-turbos | 1296 | $0.28 |
| Deepseek-v3 | 1373 | $0.28 |
| Llama-4-Maveric | 1417 | $0.82 |
| GPT-4.1 | 1366 | $8.00 |
| Grok-3-preview | 1403 | $15.00 |
| o3 by OpenAI | 1413 | $40.00 |
| Gemini-2.5-Pro | 1446 | $10.00 |

