# OpenReview forum: "Near-Optimal Online Deployment and Routing for Streaming LLMs"
_ICLR.cc/2026/Conference — ICLR 2026 Poster_

### Official Review · Reviewer_ANEN · 2025-10-26

**Soundness:** 3
**Presentation:** 3
**Contribution:** 3
**Rating:** 6
**Confidence:** 2

**Summary:**

This paper tackles the problem of online deployment and routing for streaming large language models in a realistic cloud setting where new models arrive over time and operational constraints apply. The authors formalize a novel two-timescale decision framework: at infrequent deployment stages, the system chooses which models to keep deployed (up to $M_{\max}$ models) and at the fine-grained level, each incoming query is routed to one of the deployed models to maximize quality under cost and throughput constraints. The paper proposes an algorithm called StageRoute, which (i) uses optimistic estimates of performance (UCB) and conservative estimates of cost (LCB) to select a near-optimal subset of models at each stage, incurring any one-time deployment costs, and (ii) routes each query among the chosen models by solving a linear program that respects the long-term budget and per-model rate limits. The authors prove a regret bound of $\widetilde{O}(T^{2/3})$ for StageRoute (with a matching $\widetilde{O}(T^{2/3})$ lower bound), indicating that the algorithm’s performance loss relative to an oracle is sublinear in the number of queries and essentially near-optimal for this problem setting. Empirically, StageRoute is evaluated on a benchmark (36k+ queries across 8 diverse datasets) and is shown to closely track the performance of an oracle under strict cost budgets while significantly outperforming baseline strategies, demonstrating both the practical effectiveness and robustness of the approach.

**Strengths:**

Novel Problem Formulation: The paper identifies and formalizes a unique and important problem at the intersection of LLM systems and online learning. It is, to the best of the authors’ knowledge, the first to address online LLM deployment with streaming model arrivals while accounting for a hard concurrency cap on active models, one-time deployment costs, per-model throughput limits, and a global cost budget. This formulation fills a clear gap in prior work and it mirrors real-world constraints.

Well-Designed StageRoute Algorithm: The proposed StageRoute algorithm elegantly mirrors the two-level structure of the problem. It combines strategic exploration-exploitation at deployment times with tactical optimal routing for each query. The deployment phase uses upper-confidence bounds on reward and lower-confidence bounds on cost to select a set of up to $M_{\max}$ models, ensuring a principled trade-off between trying new models and sticking with reliable ones under uncertainty. Then, within each stage, the routing phase solves a linear programming subproblem to dispatch queries in a way that maximizes expected quality while respecting the budget and per-model throughput caps. This hierarchical approach is conceptually sound and modular – for instance, the routing step could incorporate contextual features if available, without changing the deployment step. Overall, the algorithmic design addresses the identified challenges in a logical and effective manner.

Theoretical Rigor: A key strength is the thorough theoretical analysis. The authors establish an $\widetilde{O}(T^{2/3})$ regret bound for StageRoute and also prove a matching lower bound for any policy in this setting, implying that StageRoute achieves essentially the best possible learning rate for the given problem. This is a non-trivial result, as the problem’s structure (staged decisions, streaming new arms, budget and capacity constraints) is more complex than standard bandits. The regret decomposition provides insight into the costs of limited concurrency and delayed updates. Such theoretical guarantees lend strong support to the algorithm’s reliability and novelty. It’s commendable that the paper not only proposes a solution but also proves its near-optimality, meeting a high bar of technical depth.

Clarity and Presentation: The paper is clearly written and well-organized, which aids in understanding a complex problem. The authors provide a helpful conceptual diagram (Figure 1) and use it to explain the StageRoute workflow intuitively. The constraints and assumptions are well-motivated by real-world examples. Each component of the solution is described with sufficient detail, and important aspects like the modularity of the design are highlighted (e.g., noting that contextual routing or other extensions can be plugged in). The combination of theoretical and empirical sections is balanced, and the contributions are explicitly itemized early on. This clarity ensures that readers can follow the motivation, the algorithm, and the significance of results without confusion.

**Weaknesses:**

Computational Complexity and Scalability: A potential concern is the computational overhead of StageRoute’s deployment phase. Selecting the active model set is formulated as an optimization problem with combinatorial constraints (respecting the $M_{\max}$ cap, budget, etc.), which the authors solve using a mixed-integer programming (MIP) solver in their experiments. This approach may not scale well if the number of candidate models grows large or if deployment decisions need to be made very frequently. The paper does not report runtime or complexity analysis for this MIP step, so it’s unclear how feasible StageRoute would be in real-time or large-scale deployments without further engineering. While throughput limits help throttle query routing, the stage-wise optimization might become a bottleneck as the system scales, and discussing this would strengthen the paper.

Baseline Limitations: Given the novelty of the problem setting, the baselines used for comparison are relatively simple. The paper compares StageRoute mainly against a greedy heuristic (select top-$M_{\max}$ models by a UCB/LCB-derived utility) and a uniform random strategy, plus an oracle upper-bound. These are sensible reference points, but one might wonder how StageRoute fares against more sophisticated alternatives. For example, adapting a bandits with knapsacks approach or a standard multi-armed bandit algorithm with budget constraints could be attempted as a baseline. The absence of comparisons to any prior bandit or online learning algorithms makes it harder to pinpoint how much improvement comes from each novel aspect of StageRoute.

Assumption of Immediate Reward Feedback: The framework assumes that after each query is routed to a model, the system obtains a reward/quality score for that model’s answer, which is used to update the UCB values. In the experiments, this is achieved by using a benchmark dataset where each query has pre-assessed scores for each model’s response. However, in a real deployment of LLMs serving live user queries, obtaining a numerical reward signal for each response can be challenging – user feedback may be implicit, delayed, or noisy. The paper does not discuss how robust the algorithm is to noisy feedback or how it might work with proxy metrics (like user click-through or satisfaction ratings). This is a potential gap between the experimental setup and real-world application. If the reward signal is sparse or delayed, the effectiveness of the UCB-based exploration could be affected. Clarifying this assumption and its impact would be helpful, as the need for a reliable performance metric per query could limit applicability in practice.

Scope of Evaluation and Real-World Factors: It would strengthen the paper to see evaluation under a wider range of conditions. For instance, how would StageRoute perform with a larger pool of models or more frequent model introductions? Similarly, the study focuses on cost and throughput constraints, but other practical considerations like variable latency or model reliability are not deeply explored. The concurrency cap $M_{\max}$ is fixed, and it’s not fully clear how sensitive the system is to this parameter beyond what's in the theoretical bound.

**Questions:**

Choosing the Update Interval: What guidance can the authors provide on setting the stage update frequency (interval length)? There is an inherent trade-off: updating deployments more frequently allows faster adaptation to new models but also incurs more frequent deployment costs and reduces per-model learning time, whereas longer stages mean more stability but risk sticking with suboptimal models. It would be useful to discuss how an operator might choose the update schedule in practice, and whether StageRoute’s performance degrades if this parameter is not well-tuned.

Incorporating Contextual Information: The paper notes that the routing step can incorporate contextual estimators when features are available. Did the authors explore any form of contextual routing in their experiments, or could they provide examples of what contextual features might be useful? For instance, one could imagine features derived from the query to predict which model is likely to perform best. If context was not used in the current work, do the authors anticipate that adding context would further improve performance, and would any changes be needed in the theoretical analysis to accommodate contextual bandit estimators?

Deployment Cost Parameter: The formulation includes a cost for deploying a model, presumably to discourage constantly swapping models in and out. However, the paper does not detail the values or impact of these deployment costs. Clarification on how deployment costs were set and whether they had any noticeable effect on the decisions would be helpful.

Adaptation to New Models: How quickly and reliably does StageRoute adapt to a newly arrived model that has superior performance? Based on the description, the algorithm will explore new models via UCB if they appear promising, but the stage commitment means a wrong decision could delay using a good model for an entire stage. The heatmap comparison suggests StageRoute’s selection probabilities eventually align with the oracle’s optimal set. It would be insightful if the authors could elaborate on this: for example, do they observe that StageRoute sometimes sticks with an incumbent model for a stage or two before “realizing” a newcomer is better, or does the algorithm typically catch on almost immediately at the next update? Understanding this dynamic would shed light on the practical behavior of the system in a continuously evolving model pool.

---

> ### Author Response · Authors · 2025-11-20
>
> Thank you very much for your constructive comments, as well as giving the positive rating of our work. Here we would like to address the reviewer's concerns and hope that can help raise the rating of our paper.
>
> **Weakness #1: Computational Complexity and Scalability.**
>
> **Our Response:** Thank you for your question. Our two-stage design keeps optimization overhead small:
>
> **1. Deployment solve (infrequent, MIP).** For the deployment phase, Eq. (3) is a MIP, which is solved once per state (i.e., at maintenance windows), not per query. In practice, $|\mathcal{M}\_{\tau_k}|$ is modest (tens) and $M_{\max}$ is small (often single-low-double digits), so the MIP is tiny. Given this small scale, standard solver presolve routines typically yield sub-second solves even from scratch.
>
> **2. Routing solve (per query, tiny LP).** Between updates, we solve a small LP over the currently deployed models (size $\le M_{\max}$). This LP involves a budget constraint alongside standard simplex and box constraints, which admits a solver-free implementation. In our experiments using a standard solver, this takes milliseconds on a CPU.
>
> **3. How often and how fast?**
>
> *   Deployment MIP: once per stage (operator-chosen cadence).
> *   Routing LP: every query, but with the closed-form routine above. In our experiments ($T = 36,497$ queries on an Intel i9-12900HX), the entire run finished in $<10$ minutes, with per-query routing taking milliseconds and stage MIPs negligible at this scale.
>
> In practice, not too many models are deployed simultaneously, most LLM API services from providers offer on the order of tens of models, so its solving speed will be very fast. Therefore, using either Gurobi or an open-source solver (e.g., CBC, GLPK, HiGHS) is feasible.
>
> **4. Solution reuse/warm-starts.**
> As for solution reuse, it is indeed highly effective because parameters change only slightly between runs. For instance, after a routing decision, only the statistics for the selected model are updated. Thus, we can warm-start the routing LP from the previous basis or solution. Similarly, between stages, we can warm-start the deployment MIP by providing the prior active set as an initial feasible solution (MIP start), since the candidate pool changes minimally. Such warm-starting capabilities are also supported by modern open-source solvers (e.g., HiGHS). While we did not strictly require this optimization due to the fast solve times, our framework is highly amenable to these standard acceleration techniques.
>
> **Update in the Manuscript:** We have incorporated this discussion into a new paragraph titled “Computational Overhead” in Section 5 (Page 8), which is highlighted in blue.

---

> ### Author Response · Authors · 2025-11-20
>
> **Weakness #2: Baseline Limitations.**
>
> **Our Response:** Thank you for your suggestion.
>
> **Why Standard MAB Algorithms Fail Here:**
>
> Our work addresses a novel setting for online LLM deployment and routing. We acknowledge the vast and important literature on bandits, including (i) Bandits with Knapsacks (BwK) or linear contextual variants like LinUCB, which address decision-making under constraints, have indeed been applied to routing tasks; (ii) Combinatorial Multi-Armed Bandits (CMAB), which select a subset of arms in each round, address super-arm selection; and (iii) Streaming/non-stationary bandits, which address decision-making under changing environments.
>
> However, they cannot handle our setting for the following reasons:
>
> *   **Dynamic Model Pool & Concurrency Limits:** Our core challenge is managing **the streaming arrivals** of LLM models where new arms arrive over time, all under **a hard concurrency cap** (i.e., a strict limit $M_{\max}$) on the number of concurrently active arms, which admitting a newcomer forces eviction and that commitment persists for an entire stage. In contrast, existing CMAB literature often assumes a fixed base-arm set and per-round superarm selection, BwK literature handles consumable budgets but not a cardinality/support constraint on the optimal solution of a stage-level LP nor one-time deployment costs with stage commitment, and streaming bandits typically control memory/attention but don’t couple stage-wise support selection with per-query constrained routing while non-stationary bandits assume gradual drift rather than arrival-driven frontier shifts. All this literature lacks the mechanism to handle streaming arrivals or the crucial "deploy vs. drop" decision that arises from the concurrency limit.
> *   **Staged Commitment & Hierarchical Structure:** Our setting couples two timescales: a slower deployment process that decides which models remain live, and a faster per-query routing process that operates only over the deployed set. Deployment choices constrain routing; in turn, routing feedback informs the next deployment update. Standard bandit frameworks that make per-round decisions over a fixed action set do not capture the long-term, irreversible nature of these commit-and-evict choices or this hierarchical two-timescale structure.
>
> In summary, this stateful, hierarchical structure falls beyond the scope of traditional bandit formulations. We have detailed these distinctions in several places in our paper: the "Algorithmic Innovations" paragraph at the end of Section 3 explains why a new algorithm is necessary; the "Why existing analyses do not apply" paragraph in the middle of Section 4 details the theoretical novelty; and Appendix A.2 provides a comprehensive comparison with related bandit families.
>
> **Rationale for Baselines:**
>
> Given these structural differences, directly applying existing bandits algorithms is not feasible without significant, non-trivial modifications that would essentially create new algorithms, making a fair comparison difficult. Therefore, we designed the strongest possible heuristics for the deployment selection phase:
>
> 1.  **Oracle:** We emphasize that our experimental regret is derived by comparing our method to the optimal policy with full information (the Oracle). The resulting low regret serves as proof of our algorithm's optimality. Additionally, Figures 2 and 4 show that, in terms of both behavioral heatmaps and performance-cost evolution, our algorithm behaves very similarly to the optimal strategy.
> 2.  **Resource-Aware Greedy:** This baseline calculates the optimistic utility (UCB-Reward / LCB-Cost) for all models and selects the top-$M_{\max}$. It incorporates **exploration** (via UCB) and **budget awareness** (via Cost ratio). This is a powerful and widely-used heuristic in resource allocation problems adapted to our constrained setting, as most practical routing systems rely on similar utility scores.
> 3.  **Random:** This serves as a sanity check and establishes a clear lower bound on performance.
>
> To ensure a fair comparison, both Resource-Aware Greedy and Random utilize our `RouteOPT` (LP) for the routing phase. This isolates the gain specifically to `StageRoute`'s novel deployment strategy and the benefits derived from the coupled learning of the two stages. We believe these represent the most relevant and informative comparisons in the absence of directly applicable prior methods.

---

> ### Author Response · Authors · 2025-11-20
>
> **Weakness #3: Assumption of Immediate Reward Feedback.**
>
> **Our Response:**
> Our framework is more robust to noisy and delayed feedback than it may appear:
>
> 1.  **Robustness to Noise:** The UCB/LCB confidence bounds at the heart of our algorithm are inherently designed for stochastic (i.e., noisy) feedback. They work by maintaining statistical estimates (means and variances), which naturally average out noise over time.
> 2.  **Resilience to Delay:** The stage-based nature of our algorithm makes it naturally resilient to within-stage feedback delays. As long as feedback (even if delayed) arrives and is incorporated into the model statistics before the next deployment decision at the start of the next stage, the algorithm can effectively learn. The performance within one stage might be affected by very long delays, but the strategic learning over stages remains robust.
>
> ***
>
> **Weakness #4: Scope of Evaluation and Real-World Factors.**
>
> **Our Response:** Regarding the sensitivity to the concurrency cap $M_{max}$, we would like to gently point the reviewer to Figures 3, 5(a), and 5(b) in our paper, where we present a detailed sensitivity analysis for $M_{max}$ values of 3, 5 (default), and 10.
>
> ***
>
> **Question #1: Choosing the Update Interval.**
>
> **Our Response:** Our work provides both theoretical and empirical guidance:
>
> 1.  **Theoretical Guidance:** As detailed in Theorem 1 and the "Practical guidance" paragraph (lines 316-323), our theory suggests setting the number of stages $K$ to be approximately $T^{1/3}$, which translates to a stage length of $T^{2/3}$.
> 2.  **Empirical Validation:** Our experiments strongly corroborate this. As shown in Figures 3, 5(c), and 5(d), the empirically best performance is achieved with an update interval of 1000 rounds ($K \approx 36$). For our horizon $T$, the theory suggests $K \approx 33$. This remarkable alignment demonstrates that our theory provides actionable, real-world guidance for tuning this hyperparameter.
>
> ***
>
> **Question #2: Incorporating Contextual Information.**
>
> **Our Response:** Thank you for this insightful question. In our current work, we focused on the query-independent setting to establish foundational theoretical guarantees for the novel "deploy-then-route" problem. Therefore, we did not implement a contextual version in our experiments. Additionally, our framework is inherently modular: the tactical `RouteOPT` component can be readily replaced with a contextual bandit policy if reliable features are available. This highlights the extensibility of our work.
>
> ***
>
> **Question #3: Deployment Cost Parameter.**
>
> **Our Response:** Thank you for the clarifying question. In our current formulation, the primary "deployment cost" is captured implicitly by the hard concurrency cap $M_{max}$. This represents the scarcity of operational slots, which is the most dominant cost and constraint in real-world systems (e.g., limited GPU memory or API quotas). We did not model an additional per-swap numerical cost, as the concurrency limit is the binding factor.
>
> ***
>
> **Question #4: Adaptation to New Models.**
>
> **Our Response:** Thanks for your suggestion. `DeployOPT` ensures that new, unexplored models are treated optimistically and are given a chance to be deployed. The stage commitment means there is a potential delay, but this is precisely the trade-off our algorithm is designed to manage.
>
> Our empirical results in Figure 2 (the decision heatmap) provide a clear answer to how quickly StageRoute adapts. The horizontal axis represents deployment intervals (stages). When a new, superior model becomes available, we can see that StageRoute's selection probability for that model (darker color) ramps up very quickly, often in the very next stage after its arrival.
> Its behavior closely mirrors that of the Oracle (Figure 2b), which has full information. This shows that the algorithm does not get stuck on incumbents for long; the optimistic exploration mechanism is effective at identifying and promoting promising newcomers at the next available opportunity.
>
> In summary, our experiments show that StageRoute is highly effective and rapid in adapting to the evolving model landscape.

---

> ### Author Response · Authors · 2025-11-24
> **Follow-up for Reviewer ANEN**
>
> We appreciate your detailed review and the constructive suggestions. Our Nov 19 rebuttal responds to the points you flagged. If any part would benefit from further clarification, we’re glad to elaborate. If the replies addressed your concerns, we’d be grateful for your consideration during the final discussion. Thank you!

---

### Official Review · Reviewer_G3fm · 2025-10-27

**Soundness:** 2
**Presentation:** 3
**Contribution:** 3
**Rating:** 6
**Confidence:** 2

**Summary:**

The authors introduce *StageRoute* an algorithm for LLM routing that accounts for a continually updating family of models (reflecting how LLM routing must be done in practice). The authors provide theoretical guarantees for their method and an experimental analysis performed on Routerbench.

**Strengths:**

* Adapting LLM routing to an ever-changing landscape of models is a significant contribution.
* StageRoute includes a minimax optimality guarantee that also provides practical guidance on the selection of $K$ (the number of model deployment stages).

**Weaknesses:**

* The theoretical guarantees seem to require that $r_t$ and $c_t$ are drawn from model-dependendent but *query* independent distributions. It is unclear to me that this assumption makes sense in the context of studying *per-query* routing; in fact if this does hold why do any per-query routing at all?
* The empirical base lines comparisons seem lacking, the authors only compare to a greedy and random baseline, but no prior routing methods.

**Questions:**

* Can the authors add more extensive empirical comparisons to prior work on routing?
* Additionally, the authors might consider adding a study on the choice of $K$, since this seems to be the main guidance the theory provides.

---

> ### Author Response · Authors · 2025-11-20
>
> Thank you very much for your constructive comments, as well as giving the positive rating of our work. Here we would like to address the reviewer's concerns and hope that can help raise the rating of our paper.
>
> **Weakness #1: The theoretical guarantees seem to require that $r_t$ and $c_t$ are drawn from model-dependendent but query independent distributions. It is unclear to me that this assumption makes sense in the context of studying per-query routing; in fact if this does hold why do any per-query routing at all?**
>
> **Our Response:**
>
> Thank you for the thoughtful question.
>
> **Why We Focus on the Query-Independent Setting**
>
> As you noted, routing strategies are either query-dependent or query-independent. Query-dependent methods can be effective but introduce challenges: (1) They rely on predictors of per-query reward and cost, so decisions are highly sensitive to prediction error, especially under multiple constraints. Their performance can also drift over time, which is problematic in our online setting; and (2) Training typically requires full-feedback datasets in which each query is evaluated by all candidate LLMs. Given the rapid pace of model releases, maintaining such comprehensive, up-to-date datasets is impractical.
>
> In our analysis we assume model-dependent, query-agnostic rewards and costs: for a fixed model $m$, $(r_t,c_t)$ are i.i.d. across arrivals with means $(\mu_m,c_m)$. This captures production settings where *the mix of query types is approximately stationary over time*. This assumption is used solely for analysis and does not constrain the algorithm. The resulting worst-case guarantees do not depend on the availability or quality of query features, nor on pre-trained predictors.
>
> **The Necessity of Per-Query Routing**
>
> Even under this assumption, per-query routing remains necessary:
>
> *   *Implementing the optimal mixture under constraints.* With known $(\mu_m,c_m)$, the stage-optimal policy is generally a randomized mix over deployed models (to satisfy the per-query capacity limits and the long-run cost budget). That mix must be implemented on individual arrivals, i.e., per-query routing is how the mixture is realized.
> *   *Learning and adaptation.* Means and costs are unknown and updated online, and newly arriving models change the feasible set. We therefore recompute the routing distribution as estimates tighten and the active set evolves, even in a query-agnostic world.
> *   *Constraint tracking in real time.* Capacity constraints are per-query, and budget control uses conservative cost estimates at the query timescale. A static "once-per-stage" choice can violate these constraints when realized costs deviate. In contrast, per-query routing keeps the system within limits.
>
> **Framework Extensibility**
>
> Finally, our StageRoute framework is inherently modular. Although our theory is established in a query-independent setting, the RouteOPT component can be easily replaced with other algorithms if query features are available, while the concept of the DeployOPT layer would remain unchanged. This highlights the extensibility of our framework.

---

> ### Author Response · Authors · 2025-11-20
>
> **Weakness #2 & Questions #1: The empirical base lines comparisons seem lacking, the authors only compare to a greedy and random baseline, but no prior routing methods. Can the authors add more extensive empirical comparisons to prior work on routing?**
>
> **Our Response:**
>
> Thank you for your question. There is a rich body of work on LLM routing. We'd like to clarify why a direct comparison is not applicable in our novel "deploy-then-route" setting.
>
> **Core Mismatch with Prior Routing Methods:**
>
> Prior routing methods operate on a crucial assumption: the set of available LLMs is static and given. Their task is to select the best model from this pre-existing, often large, pool for a given query.
> In contrast, our work addresses the preceding, more fundamental operational problem: which small subset of models ($M_{\max}$) should be deployed and active in the first place? This is especially challenging in a dynamic environment with streaming model arrivals and a hard concurrency cap.
>
> This makes a direct comparison an "apples-to-oranges" problem. Prior routing methods lack a "deployment policy". They are not designed to handle our core constraints (streaming arrivals, concurrency cap, staged commitment) and thus cannot be applied off-the-shelf to our problem.
>
> **Rationale for Baselines:**
>
> Since our primary contribution is the novel, stage-wise deployment strategy, our experiments are designed to rigorously evaluate precisely this component. Therefore, we designed the strongest possible heuristics for the deployment selection phase:
>
> 1.  **Oracle:** We emphasize that our experimental regret is derived by comparing our method to the optimal policy with full information (the Oracle). The resulting low regret serves as proof of our algorithm's optimality. Additionally, Figures 2 and 4 show that, in terms of both behavioral heatmaps and performance-cost evolution, our algorithm behaves very similarly to the optimal strategy.
> 2.  **Resource-Aware Greedy:** This baseline calculates the optimistic utility (UCB-Reward / LCB-Cost) for all models and selects the top-$M_{\max}$. It incorporates **exploration** (via UCB) and **budget awareness** (via Cost ratio). This is a powerful and widely-used heuristic in resource allocation problems adapted to our constrained setting, as most practical routing systems rely on similar utility scores.
> 3.  **Random:** This serves as a sanity check and establishes a clear lower bound on performance.
>
> To ensure a fair comparison, both Resource-Aware Greedy and Random utilize our `RouteOPT` (LP) for the routing phase. This isolates the gain specifically to `StageRoute`'s novel deployment strategy and the benefits derived from the coupled learning of the two stages. We believe these represent the most relevant and informative comparisons in the absence of directly applicable prior methods.
>
> In summary, we did not include prior routing methods as baselines because they are structurally incompatible with our problem setting. Instead, our evaluation provides a clear and fair assessment of our core contribution by comparing it against strong heuristics and the optimum.
>
> ***
>
> **Questions #2: Additionally, the authors might consider adding a study on the choice of $K$, since this seems to be the main guidance the theory provides.**
>
> **Our Response:**
>
> Thank you for your suggestion. We have already conducted relevant experiments and compared the results. Figures 3, 5(c), and 5(d) in the paper show the change in regret resulting from different choices of $K$ (which corresponds to different update intervals, as the total number of rounds is the product of $K$ and the update interval). It can be seen that the regret is minimized when the update interval is 1000, at which point $K$ is approximately 36. Meanwhile, our theoretical analysis suggests a $K$ of $T^{1/3}$, which is approximately 33, and a corresponding update interval of about 1100. As you can see, these values are very close, indicating that our experimental results and theoretical analysis corroborate each other.

---

> ### Author Response · Authors · 2025-11-24
> **Follow-up for Reviewer G3fm**
>
> Thank you again for your thoughtful review. We posted our rebuttal on Nov 19 addressing the points you raised. If anything remains unclear, we’re happy to clarify. If our responses resolved your concerns, we’d appreciate your consideration in the final discussion. Thanks again for your time and feedback.

---

### Official Review · Reviewer_bJS8 · 2025-11-01

**Soundness:** 3
**Presentation:** 3
**Contribution:** 3
**Rating:** 6
**Confidence:** 3

**Summary:**

This paper studies the problem of managing a streaming inventory of LLMs under concurrency and per-query cost constraints. The authors present StageRoute, a hierarchical online decision-making framework for adaptive, cost-aware, and scalable deployment and routing of large language models at scale. The algorithm achieves a regret bound of $\tilde{O}(T^{2/3})$ and a matching lower bound, establishing near-optimality. Empirically, StageRoute tracks a strong oracle under tight budget constraints across diverse workloads.

**Strengths:**

1. The paper addresses a timely and practical challenge in LLM serving systems, where cost and scalability are key bottlenecks.
2. The theoretical analysis is rigorous and clarifies the trade-off between adaptivity and learnability.
3. The presentation and organization have improved since the NeurIPS version, with better discussion of parameter dependencies and clearer empirical exposition.

**Weaknesses:**

*Disclosure:* I also reviewed this paper in its earlier NeurIPS submission. Compared with that version, I find that the current paper has made several substantial improvements:

1. The authors now clearly explain why the dependence on $K$ becomes invalid when $K \ge O(T^{1/3})$, addressing my previous concern both theoretically (line 317) and empirically (line 466).
2. While the algorithmic design still combines several existing ideas, the paper now provides a more thoughtful discussion of how these perspectives fit together, making the unification more convincing.
3. Although the experimental setup still involves only two relatively simple baselines, the added analysis clarifies why StageRoute can approach the optimal solution, which mitigates the earlier concern.

--------------------------previous review------------------------------------------
1. **Regret Upper Bound Interpretation**: The regret bound depends on K\sqrt{K}, where KK is the number of deployment stages. While this appears reasonable under a fixed-stage model, it becomes problematic in fully adaptive environments where $K=T$. In that case, the regret becomes $O(T)$, which suggests linear regret—a degenerate and counterintuitive result. One would expect more adaptivity (i.e., frequent updates) to improve performance, not worsen it. Clarifying this behavior or bounding regret in fully adaptive settings would strengthen the theoretical contribution.
2. **Algorithmic Novelty**: Although the problem is framed as hierarchical, the optimization subproblems in both stages reduce to similar formulations (e.g., Eq. (7)), and the algorithmic components largely resemble standard knapsack or budgeted bandit methods (e.g., [4]). It would be beneficial for the authors to clarify what aspects are truly novel beyond the hierarchical decomposition, such as any new learning techniques, budget handling strategies, or theoretical proof techniques introduced specifically for this setting.
3. **Limited Empirical Comparisons**: The experimental section only compares StageRoute against two simple baselines. This is a missed opportunity, given the availability of related work on budgeted bandits (e.g., [4, 8, 19, 22]) and LLM routing (e.g., [13, 25, 27, 36]). Including stronger baselines from these domains would provide a more convincing evaluation of StageRoute’s practical advantages and generality.

**Questions:**

I do not have major concerns for this version.

---

> ### Author Response · Authors · 2025-11-20
>
> Thank you very much for the careful re-review and for noting the improvements over the NeurIPS version. We’re glad the current draft addresses your earlier concerns (e.g., the dependence on $K$ and how StageRoute approaches the oracle). We appreciate your positive assessment and are happy to clarify anything else.

---

### Official Review · Reviewer_DHJM · 2025-11-12

**Soundness:** 3
**Presentation:** 4
**Contribution:** 3
**Rating:** 4
**Confidence:** 3

**Summary:**

The authors of this paper look at a new-ish problem - which models should be deployed given a tight budget. They introduce a new method called StageRoute, which allows incoming query to be routed to an ever-changing set of models.  The authors prove some regret bounds, establish near-optimalty and also provide experiments.

**Strengths:**

1. The paper looks at a very practical problem, how does one choose which models to deploy in a tight setup? This is very useful for practical settings
2. I carefully checked several theorems and lemmas (not all), they look correct.
3. The experiments are quite comprehensive. In figure 3, it is quite clear that cumulative regret is fairly low for StageRoute. The sensitivity analysis is quite nice too.

**Weaknesses:**

1. The paper does not do a great job of connecting with existing explore-exploit literature. Please expand upon the section why existing analyses does not apply. This bleeds into the experiments. Why compare only with greedy and random? This to me is the biggest drawback of this paper, there is a ton of literature in the MAB space that could have been used to create stronger baselines
2. In eq(3), it is important for the authors to describe the details of the problem. Is it a MIP? IP? LP? They briefly mention this in the experiments with MIP + Gurobi, a deeper discussion is warranted.
3. The main body of the paper does not really talk about the cost of running optinmization. How does this optimization scale? Can you reuse solutions from previous runs? How often is the problem solved? Not everyone has access to Gurobi. This is quite important.

**Questions:**

See weaknesses

---

> ### Author Response · Authors · 2025-11-20
>
> Thank you for your constructive comments. Here we would like to address the reviewer's concerns and hope that can help raise the rating of our paper.
>
> **Weakness #1: The paper does not do a great job of connecting with existing explore-exploit literature. Please expand upon the section why existing analyses does not apply. This bleeds into the experiments. Why compare only with greedy and random? This to me is the biggest drawback of this paper, there is a ton of literature in the MAB space that could have been used to create stronger baselines.**
>
> **Our Response:**
>
> Thank you for your suggestion.
>
> **Why Standard MAB Algorithms Fail Here:**
>
> Our work addresses a novel setting for online LLM deployment and routing. We acknowledge the vast and important literature on bandits, including (i) Bandits with Knapsacks (BwK) or linear contextual variants like LinUCB, which address decision-making under constraints, have indeed been applied to routing tasks; (ii) Combinatorial Multi-Armed Bandits (CMAB), which select a subset of arms in each round, address super-arm selection; and (iii) Streaming/non-stationary bandits, which address decision-making under changing environments.
>
> However, they cannot handle our setting for the following reasons:
>
> *   **Dynamic Model Pool & Concurrency Limits:** Our core challenge is managing **the streaming arrivals** of LLM models where new arms arrive over time, all under **a hard concurrency cap** (i.e., a strict limit $M_{\max}$) on the number of concurrently active arms, which admitting a newcomer forces eviction and that commitment persists for an entire stage. In contrast, existing CMAB literature often assumes a fixed base-arm set and per-round superarm selection, BwK literature handles consumable budgets but not a cardinality/support constraint on the optimal solution of a stage-level LP nor one-time deployment costs with stage commitment, and streaming bandits typically control memory/attention but don’t couple stage-wise support selection with per-query constrained routing while non-stationary bandits assume gradual drift rather than arrival-driven frontier shifts. All this literature lacks the mechanism to handle streaming arrivals or the crucial "deploy vs. drop" decision that arises from the concurrency limit.
> *   **Staged Commitment & Hierarchical Structure:** Our setting couples two timescales: a slower deployment process that decides which models remain live, and a faster per-query routing process that operates only over the deployed set. Deployment choices constrain routing; in turn, routing feedback informs the next deployment update. Standard bandit frameworks that make per-round decisions over a fixed action set do not capture the long-term, irreversible nature of these commit-and-evict choices or this hierarchical two-timescale structure.
>
> In summary, this stateful, hierarchical structure falls beyond the scope of traditional bandit formulations. We have detailed these distinctions in several places in our paper: the "Algorithmic Innovations" paragraph at the end of Section 3 explains why a new algorithm is necessary; the "Why existing analyses do not apply" paragraph in the middle of Section 4 details the theoretical novelty; and Appendix A.2 provides a comprehensive comparison with related bandit families.
>
> **Rationale for Baselines:**
>
> Given these structural differences, directly applying existing bandits algorithms is not feasible without significant, non-trivial modifications that would essentially create new algorithms, making a fair comparison difficult. Therefore, we designed the strongest possible heuristics for the deployment selection phase:
>
> 1.  **Oracle:** We emphasize that our experimental regret is derived by comparing our method to the optimal policy with full information (the Oracle). The resulting low regret serves as proof of our algorithm's optimality. Additionally, Figures 2 and 4 show that, in terms of both behavioral heatmaps and performance-cost evolution, our algorithm behaves very similarly to the optimal strategy.
> 2.  **Resource-Aware Greedy:** This baseline calculates the optimistic utility (UCB-Reward / LCB-Cost) for all models and selects the top-$M_{\max}$. It incorporates **exploration** (via UCB) and **budget awareness** (via Cost ratio). This is a powerful and widely-used heuristic in resource allocation problems adapted to our constrained setting, as most practical routing systems rely on similar utility scores.
> 3.  **Random:** This serves as a sanity check and establishes a clear lower bound on performance.
>
> To ensure a fair comparison, both Resource-Aware Greedy and Random utilize our `RouteOPT` (LP) for the routing phase. This isolates the gain specifically to `StageRoute`'s novel deployment strategy and the benefits derived from the coupled learning of the two stages. We believe these represent the most relevant and informative comparisons in the absence of directly applicable prior methods.

---

> ### Author Response · Authors · 2025-11-20
>
> **Weakness #2: In eq(3), it is important for the authors to describe the details of the problem. Is it a MIP? IP? LP? They briefly mention this in the experiments with MIP + Gurobi, a deeper discussion is warranted.**
>
> **Our Response:** Thank you for the suggestion. Equation (3) is a Mixed-Integer Program (MIP).
> The combinatorial part comes from the cardinality cap on the active set. A standard encoding introduces a binary "activation" variable $z_m\in\{0,1\}$ for each model and enforces $\sum_{m} z_m = \min(M_{\max},|\mathcal M_{\tau_k}|),$ where $0 \le d_m \le \alpha_m z_m$.
>
> The full stage-$k$ deployment problem is therefore:
>
> \begin{align*}
> \max_{d,z}&\sum_{m\in\mathcal M_{\tau_k}} \mu^U_m d_m,
> \quad\text{s.t.}\quad\sum_m c^L_m d_m \le b, ~\sum_m d_m = 1, ~0\le d_m \le \alpha_m z_m, ~\sum_m z_m = \min(M_{\max},|\mathcal M_{\tau_k}|), ~z_m\in\{0,1\}.
> \end{align*}
>
> The binaries $z_m$ select which models are live, while the continuous $d_m$ (bounded by capacities $\alpha_m$) encode the optimistic deployment mix used only to pick $\mathcal{D}_k$ (we do not use $d_m$ as routing probabilities later).
> This formulation is essentially the same as Equation (3) in the original paper, but it perhaps better illustrates the underlying essence.
>
> **Update in the Manuscript:** We have incorporated this discussion into a new paragraph in Section 3 (Page 5), highlighted in blue, where we (i) explicitly state after Eq. (3) that it is a MIP, and (ii) include the corresponding binary-variable formulation.
>
> ***
>
> **Weakness #3: The main body of the paper does not really talk about the cost of running optinmization. How does this optimization scale? Can you reuse solutions from previous runs? How often is the problem solved? Not everyone has access to Gurobi. This is quite important.**
>
> **Our Response:** Thank you for the question.
> Our two-stage design keeps optimization overhead small:
>
> **1. Deployment solve (infrequent, MIP).** As we responded above to your Weakness #2, Eq. (3) is a MIP, which is solved once per state (i.e., at maintenance windows), not per query. In practice, $|\mathcal{M}\_{\tau_k}| $ is modest (tens) and $M_{\max}$ is small (often single-low-double digits), so the MIP is tiny. Given this small scale, standard solver presolve routines typically yield sub-second solves even from scratch.
>
> **2. Routing solve (per query, tiny LP).** Between updates, we solve a small LP over the currently deployed models (size $\le M_{\max}$). This LP involves a budget constraint alongside standard simplex and box constraints, which admits a solver-free implementation. In our experiments using a standard solver, this takes milliseconds on a CPU.
>
> **3. How often and how fast?**
>
> *   Deployment MIP: once per stage (operator-chosen cadence).
> *   Routing LP: every query, but with the closed-form routine above. In our experiments ($T = 36,497$ queries on an Intel i9-12900HX), the entire run finished in $<10$ minutes, with per-query routing taking milliseconds and stage MIPs negligible at this scale.
>
> In practice, not too many models are deployed simultaneously, most LLM API services from providers offer on the order of tens of models, so its solving speed will be very fast. Therefore, using either Gurobi or an open-source solver (e.g., CBC, GLPK, HiGHS) is feasible.
>
> **4. Solution reuse/warm-starts.**
> As for solution reuse, it is indeed highly effective because parameters change only slightly between runs. For instance, after a routing decision, only the statistics for the selected model are updated. Thus, we can warm-start the routing LP from the previous basis or solution. Similarly, between stages, we can warm-start the deployment MIP by providing the prior active set as an initial feasible solution (MIP start), since the candidate pool changes minimally. Such warm-starting capabilities are also supported by modern open-source solvers (e.g., HiGHS). While we did not strictly require this optimization due to the fast solve times, our framework is highly amenable to these standard acceleration techniques.
>
> **Update in the Manuscript:** We have incorporated this discussion into a new paragraph titled “Computational Overhead” in Section 5 (Page 8), which is highlighted in blue.

---

> ### Author Response · Authors · 2025-11-24
> **Follow-up for Reviewer DHJM**
>
> Thank you again for your thoughtful review. Since our Nov 19 rebuttal, we wanted to surface three clarifications most relevant to your comments:
> - **Eq. (3) is an MIP.** The active-set cardinality constraint makes the problem mixed-integer. We now state this explicitly after Eq. (3) and briefly explain the combinatorial origin.
> - **Computation.** DeployOPT (MIP) is solved only at stage boundaries. RouteOPT is a small LP over at most $M_{max}$ deployed models. Our full RouterBench run ($T=36,497$) finished in under $10$ minutes on a single CPU (Intel i9-12900HX). We added a “Computational Overhead” paragraph in Section 5 (p. 8).
> - **Relation to MAB/BwK/CMAB/Streaming.** We expanded the discussion in the rebuttal to detail why prior analyses do not apply in our setting (dynamic arrivals, stage-level commitment, and a live-set cap). Those discussions could also be found in the original submission.
>
> If anything remains unclear, we are happy to clarify. If these points address your concerns, we would be grateful if you would consider updating your evaluation. Thank you again for your time and constructive feedback.

---

### Author Response · Authors · 2025-12-02
**Author AC Comment: Rebuttal Summary**

Dear AC,

We sincerely appreciate your efforts in handling our submission and thank you for your time and dedication throughout this process. Since reviews were reverted to pre-discussion and reviewers can no longer reply, we’re sharing a concise summary of how the main concerns were addressed to help your decision.

**1. Highlights & Consensus**

*   All four reviewers recognized the practical importance of the problem (dynamic LLM deployment under budgets/throughput).
*   Theoretical analysis was viewed as sound and practically informative. Experiments were found comprehensive with low regret and solid sensitivity analysis.
*   Three reviewers scored the paper 6 (marginally above threshold). The remaining reviewer (DHJM) scored 4 but marked Soundness/contribution = “Good”, and Presentation = “Excellent”.

**2. Key clarifications since initial reviews**

*   **Relation to MAB/BwK/CMAB/Streaming (reviewer DHJM):** Our setting entails *a hard concurrency cap* and *streaming arrivals*, forcing irreversible “deploy vs. drop” decisions at stage boundaries. This staged, two-timescale structure with support constraints is not handled by standard MAB families. We provided a detailed comparison with variants such as Bandits with Knapsacks, linear contextual bandits (e.g., LinUCB), CMAB, and streaming/non-stationary bandits, explicitly demonstrating why they are unsuitable for our setting. Pointers already in the paper: Sec. 3 “Algorithmic Innovations” (why a new algorithm), Sec. 4 “Why existing analyses do not apply” (theory), and App. A.2 (comparative discussion).
*   **Formulation of Eq. (3) (reviewer DHJM):** We explicitly state it is a mixed-integer program (MIP). The combinatorics come from the active-set (cardinality) constraint.
*   **Computation (reviewers DHJM and ANEN):** Deployment (DeployOPT, MIP) runs only at stage boundaries; routing (RouteOPT) is a small LP over at most $M_{\max}$ models per query. Our full RouterBench run with $T=36,497$ queries completed in under 10 minutes on a single CPU. We added a short “Computational Overhead” paragraph (frequency, scalability, and warm-starts).
*   **Baselines & comparability to prior routers (Reviewers G3fm and ANEN):** Most prior LLM routers assume *a fixed, already-deployed* pool and optimize per-query selection; they do not make deployment decisions under a concurrency cap or handle streaming arrivals. A fair comparison would require grafting a deployment policy onto each router; the natural choice reduces to our resource-aware greedy deployment baseline. To isolate our novel piece (deployment under caps/arrivals), we hold routing fixed via RouteOPT and compare against Oracle/Resource-Aware Greedy/Random.

**3. Additional positive signals**

*   **Reviewer bJS8 (score 6):** “Compared with the earlier NeurIPS submission, the current paper has made several substantial improvements; I do not have major concerns for this version.”

We hope this summary is helpful in facilitating your decision. Once again, we thank you and all the reviewers for your thoughtful evaluation and for helping us improve this work.

Sincerely,

Authors of Paper #1169

---

### Meta-Review · Area_Chair_ByBH · 2025-12-30

**Summary:**

The paper addresses the challenge of dynamically deploying large language models (LLMs) under budget and capacity constraints as new models continuously arrive. StageRoute employs a two-stage approach:

Deployment Stage: Determines which models to keep active using optimistic reward upper-confidence and conservative cost lower-confidence bounds.
Routing Stage: Routes incoming queries to the selected models by solving a budget- and throughput-constrained bandit problem.
The authors demonstrate near-optimal regret bounds for StageRoute, validated through experiments showing its superior performance against simple baselines, thereby establishing its relevance in the context of rapidly evolving LLMs.

The main concerns of the reviewers are as follows:

Literature Connection: Reviewers indicated that the paper could improve its connection to existing multi-armed bandit (MAB) frameworks and other similar methods.
Baseline Comparisons: Reviewers expressed that the chosen baselines for comparison were too simplistic and did not effectively represent the current state-of-the-art.
Computational Complexity: Questions arose regarding the scalability and efficiency of the algorithm, especially concerning the mixed-integer programming (MIP) used for model selection.
Feedback Mechanism: Assumptions regarding immediate feedback from model responses were questioned, particularly how realistic this is in practical deployments.

**Reviewer Concerns:**

Connection to Literature: The authors expanded on how existing MAB approaches, such as Bandits with Knapsacks (BwK) and Combinatorial Multi-Armed Bandits (CMAB), do not apply to their two-timescale decision framework, clarifying the necessity of their proposed model. I am convinced by the authors' responses here.

Baseline Comparisons: While justifying the use of basic heuristics, the authors provided extensive insights into why more complex methods were not suitable for direct comparison. They indicated that prior routers do not account for the critical deployment decisions needed in their novel setting.

Computational Complexity: The authors detailed the efficiency of the algorithm, highlighting that the MIP is solved infrequently (only at maintenance windows), and demonstrated that the overall computational cost remains low even with the current framework, addressing potential scalability issues.

Feedback Mechanism: They clarified that the algorithm remains robust to noisy and delayed feedback, enabled by its design that maintains statistical estimates of rewards and costs, thus allowing effective learning and adaptation even with imperfect information.

**Reviewer Scores:**

Most of the scores support acceptance of the paper. The rebuttal by the authors is also convincing. I also support acceptance of this paper as a poster for the following reasons.

Firstly, the paper tackles a highly relevant issue in managing LLMs amid constant change, directly addressing the needs of modern AI deployment scenarios, making a significant contribution to the field. Secondly, the establishment of near-optimal regret bounds adds a strong theoretical foundation, presenting a valuable contribution that enhances understanding in operational decision-making under constraints. Thirdly, the experimental evaluation provides robust evidence that StageRoute outperforms existing approaches under tight constraints, demonstrating practical applicability and effectiveness. Finally, the paper is well-structured and clearly articulates the algorithm's workflow, contributing to the overall quality of communication and understanding of complex concepts.

---

### Decision · Program_Chairs · 2026-01-26

Accept (Poster)